# Can LLMs Refuse Questions They Do Not Know? Measuring Knowledge-Aware Refusal in Factual Tasks

**Wenbo Pan**[1]    **Jie Xu**[1]    **Qiguang Chen**[2]    **Junhao Dong**[3]
**Libo Qin**[4]    **Xinfeng Li**[3*]    **Haining Yu**[2]    **Xiaohua Jia**[1]
[1]Department of Computer Science, City University of Hong Kong, Hong Kong SAR, China
[2]Harbin Institute of Technology, Harbin, China
[3]College of Computing and Data Science, Nanyang Technological University, Singapore
[4]School of Computer Science and Engineering, Central South University, Changsha, China

## Abstract

Large Language Models (LLMs) should refuse to answer questions beyond their knowledge. This capability, which we term *knowledge-aware refusal*, is crucial for factual reliability, while existing metrics fail to capture this ability. In this work, we propose the *Refusal Index (RI)*, a novel and principled metric that measures how accurately LLMs refuse questions they do not know. We define RI as Spearman's rank correlation between refusal probability and error probability. RI is practically measurable with a lightweight two-pass evaluation method which only require observed refusal rates across two standard evaluation runs. Extensive experiments across 16 models and 5 datasets demonstrate that RI accurately quantifies a model's knowledge-aware refusal capability. Notably, RI remains stable across different refusal rates and provides consistent model rankings independent of a model's overall accuracy and refusal rates. These properties suggest RI captures a stable, intrinsic aspect of model knowledge calibration. More importantly, RI provides insight into an important but previously overlooked aspect of LLM factuality: while LLMs achieve high accuracy on factual tasks, their refusal behavior can be unreliable and fragile.

## 1 Introduction

Large Language Models (LLMs) are increasingly used for knowledge-intensive factual tasks, such as long-term reasoning (Chen et al., 2025) and specialized expert domains (Wang et al., 2025; Lin et al., 2024; Mahdavi et al., 2025). Despite these capabilities, LLMs are often poorly calibrated, frequently providing incorrect answers with high confidence (Huang et al., 2025). An intuitive solution is to enable models to refuse questions beyond their knowledge (Yin et al., 2023b). Recent work has explored and strengthened this ability by inducing more accurate refusals with prompting (Cheng et al., 2024; Kadavath et al., 2022b) or fine-tuning (Zhang et al., 2024; Kapoor et al., 2024). This capability is important for making models more reliable when answering factual questions.

In this paper, we formalize this ability, an LLM's ability to refuse factual questions it does not know, as *knowledge-aware refusal*. A truly knowledge-aware refusal assesses a model's judgment in two ways: how well a model refuses questions beyond its knowledge (avoiding *overconfidence*) and how well it avoids refusing questions it would answer correctly (avoiding *over-refusal*). Traditional factuality metrics fail to capture this property, leaving knowledge-aware refusal insufficiently measured.

Motivated by this gap, we propose a metric called the *Refusal Index (RI)* to measure knowledge-aware refusal in factual tasks, which features two key properties: **(1) accurate estimates of knowledge-aware refusal**: We formally define the Refusal Index as the Spearman correlation between refusal probabilities and error probabilities (Section 3). This definition is independent of refusal rate and directly targets refusal behavior, making it an unbiased measure. **(2) lightweight evaluation**

---

*Corresponding author: Xinfeng Li (`lxfmakeit@gmail.com`)

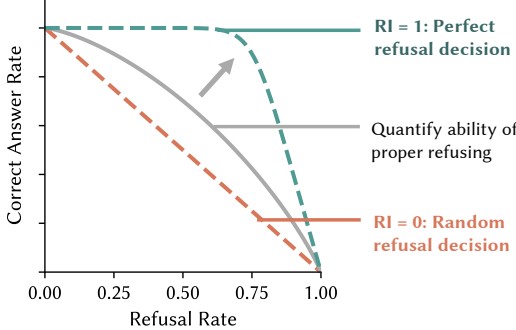 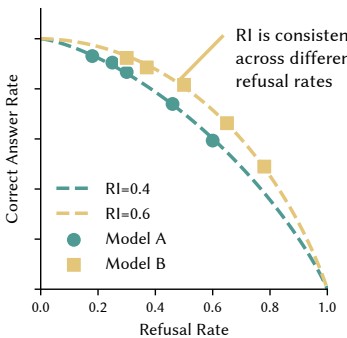

Figure 1: Illustration of Refusal Index (RI) on the accuracy-refusal trade-off, where making more refusals drops total number of correct answers. Left: Refusal Index models how the correct answer rate drops with increasing refusal rate. Right: Empirical correct answer rates for the same model at different refusal rates align with the Refusal Index.

**procedure**: Unlike calibration metrics that require expensive calibration processes, RI only needs two standard evaluation passes to compute. Specifically, we first evaluate a model on a factual question-answering dataset, collecting correct answer rates and refusal rates. Then, we run a second evaluation pass to regenerate answers for refused questions. Finally, we compute RI using the correct answer rates and refusal rates from both evaluation passes.

We perform extensive experiments and analyses to validate RI across 16 models on 5 datasets. As shown in Figure 1, RI parameterizes the relationship between correct answer rates and refusal rates, with consistent RI scores on the same model at different refusal rates. We also discover that RI has high agreement with established calibration metrics and provides consistent model rankings independent of a model's correctness and refusal rates.

Beyond RI's efficacy in capturing knowledge-aware refusal, we leverage it to reveal a critical gap in current factuality evaluation: the disconnect between factual accuracy and knowledge-aware refusal capabilities. Our analysis reveals three key insights that traditional metrics overlook:

- **RI identifies persistent capability gaps.** While LLMs achieve high accuracy on factual tasks, their refusal behavior is unreliable. This gap remains stable across different prompting strategies and cannot be resolved by simply improving accuracy or adjusting refusal rates.
- **Training data and pipelines influence refusal behavior.** Model family emerges as the strongest predictor of knowledge-aware refusal ability, with certain families consistently outperforming others regardless of model scale.
- **Knowledge-aware refusal is sensitive to noisy context.** Models exhibit degraded refusal performance when ground truth information is unavailable in the provided context, suggesting over-reliance on contextual cues.

These findings demonstrate that RI captures an essential dimension of model reliability that is overlooked by existing factuality metrics, highlighting the need to incorporate knowledge-aware refusal measures for a more comprehensive factuality evaluation.

Table 1: Baseline factuality metrics used for comparison. $c$ and $r$ denote correct answer rate among all questions and refusal rate among all questions, respectively.

| Metric | Formula | Definition |
|---|---|---|
| Correct Answer Rate | $c$ | Proportion of correct answers among all questions |
| Refusal Rate | $r$ | Proportion of refusal answers among all questions |
| Correct given Attempted (C/A) | $c/(1-r)$ | Correct answer rate among answered questions |
| F-score | $2c/(2-r)$ | Harmonic mean of Correct Answer Rate and C/A |
| Weighted Score | $c - p(1-r)$ | Weighted difference of $c$ and $r$ |
| Refusal Index | Eq. (6) | Correlation between refusal and answer incorrectness |

## 2 BACKGROUND

**Knowledge-Aware Refusal.** Knowledge-aware refusal measures whether a model can appropriately decline to answer questions it doesn't know. When we define "knowing" as the ability to provide correct answers, knowledge-aware refusal capability can be quantified by the alignment between a model's refusal decisions and its answer incorrectness. This ability is fundamental for reliable deployment of LLMs in factual tasks. Previous works have explored to evaluate this capability (Cheng et al., 2024; Kapoor et al., 2024) using refusal-rate-based and calibration-based metrics, but both exhibit distinct limitations.

**Limitations of Refusal-Rate-Based Metrics.** Refusal rate alone only measures the frequency of refusals, without capturing the correlation between refusal and answer correctness. For instance, one can prompt an LLM to be more cautious, thereby increasing the refusal rate without actually improving the model's knowledge-aware refusal ability. To address this limitation, recent works have combined refusal rates with correct answer rates for evaluation (Wei et al., 2024; Bang et al., 2025). The underlying intuition is that if a model refuses more samples while maintaining its correct answer rate, it demonstrates a stronger ability to identify uncertain questions. For example, SimpleQA (Wei et al., 2024) employs an F1 score between the correct answer rate within answered questions and the refusal rate to balance over-refusal and over-confidence. We list these combined metrics in Table 1. However, such combinations of refusal rates and correct answer rates are heuristics designed to penalize over-refusal, which fail to capture the fundamental correlation between refusal and incorrectness. As our experiments demonstrate in Section 4.2, these metrics do not measure a consistent underlying ability: when prompting models to increase or decrease their refusal rates, these metric values fluctuate significantly (e.g. F1 score used in SimpleQA varies by up to 70%).

**Limitations of Calibration-Based Metrics.** Other works employ calibration methods, which estimate the uncertainty of model outputs and compute the correlation with answer correctness using metrics such as ECE and AUROC. For example, some methods (Xiong et al., 2023) instruct models to assign verbalized confidence scores to their own outputs, while others (Ulmer et al., 2024) train auxiliary models to predict confidence from text outputs. A more faithful approach involves repeatedly sampling multiple outputs for a single question and using the frequency of refusal answers as an uncertainty measure (Wei et al., 2024). However, the confidence scores derived from these methods cannot fully represent a model's refusal probability. First, studies eliciting verbalized confidence report systematic overconfidence and high ECE, while asking the same model to vote across samples (e.g., SimpleQA) yields frequency-based curves much closer to the diagonal for larger models (Xiong et al., 2023; Wei et al., 2024). Second, auxiliary calibrators such as APRICOT or rank-calibration frameworks can produce near-perfect ECE/AUROC (Ulmer et al., 2024; Huang et al., 2024), yet these numbers mainly reflect the auxiliary predictor or ranking procedure rather than the base model's refusal behavior. Third, white-box confidence proxies like P(True) can even appear well calibrated on multiple-choice settings (Kadavath et al., 2022a), further showing that calibration verdicts swing with the chosen estimator. Therefore, while these methods provide valuable insights into calibration properties within LLMs, their uncertainty estimates do not directly reflect model refusal probability. In real-world applications, we expect models to abstain from providing uncertain answers. Thus, measuring the correlation between refusal behavior and incorrectness in black-box settings remains an unsolved yet crucial challenge for assessing the factual reliability of language models. Appendix L provides an empirical comparison of three representative calibrators ($P(\text{IK})$, APRICOT, and $P(\text{Answering})$) on Qwen3-32B, showing that they disagree and that only the sampling-based method exposes the over-confidence captured by RI.

> **Properties of Effective Measurement**
>
> *We identify three key properties that an effective metric for knowledge-aware refusal should satisfy:*
>
> *1.* ***Faithful:*** *Accurately quantify knowledge-aware refusal capability.*
>
> *2.* ***Consistent:*** *Remain stable across different refusal rates induced by varying prompts or instructions.*
>
> *3.* ***Direct:*** *Derive directly from black-box LLM refusal decisions, without relying on auxiliary models.*

## 3 REFUSAL INDEX

**Scope.** Our evaluation settings follow widely used factuality evaluations like SimpleQA and Truth-fulQA (Wei et al., 2024; Lin et al., 2022), where models provide atomic answers for short-form, factual questions. Additionally, models can refuse to answer to avoid hallucination by producing outputs such as *"I don't have enough information..."*. Following SimpleQA, each model answer is classified as correct, incorrect, or refused by comparing it against the ground truth. The classification results are used to estimate the model's factuality level, or in our case, the ability to make knowledge-aware refusals. This formulation avoids subjective grading and partial correctness in LLM generation, allowing more reliable measurement.

**Notations.** Formally, we denote the LLM as $f_{\text{LM}} : \mathcal{X} \to \mathcal{Y} \cup \{\perp\}$, where $x \in \mathcal{X}$ represents the input question, $y \in \mathcal{Y}$ represents the output answer, and $\perp$ denotes refusal. For the $i$-th question $x_i$ with ground truth $y_i$ in dataset $D$, we define two indicators: $W_i = \mathbf{1}\{f_{\text{LM}}(x_i) \neq y_i\}$ for incorrect outputs (including refusals) and $R_i = \mathbf{1}\{f_{\text{LM}}(x_i) = \perp\}$ for refusal responses. We define the error probability $w_i = P(f_{\text{LM}}(x_i) \neq y_i)$ and the refusal probability $r_i = P(f_{\text{LM}}(x_i) = \perp)$. Conceptually, a model with better knowledge-aware refusal should refuse more frequently as questions become more difficult. We measure this ability with the *Refusal Index*. Inspired by rank-based calibration metrics like AUROC (Niculescu-Mizil & Caruana, 2005), we define the Refusal Index as the correlation between refusal probabilities and error probabilities:

**Definition 3.1** (Refusal Index)**.** Refusal Index $\rho_S$ is defined as Spearman's rank correlation between the model's refusal probability $r_i$ and the error probability $w_i$ as follows:

$$\text{Refusal Index} = \rho_S = \text{Corr}(\text{Rank}(r_i), \text{Rank}(w_i)). \tag{1}$$

The intuition behind the definition is that a model achieves *perfect knowledge-awareness* when its refusal probability increases monotonically with error probability, making it more likely to refuse as questions become more difficult:

$$w_i \leq w_j \iff P(f_{\text{LM}}(x_i) = \perp) \leq P(f_{\text{LM}}(x_j) = \perp). \tag{2}$$

Note that this differs from error-based calibration metrics like Expected Calibration Error (ECE), which quantify absolute discrepancies between $r_i$ and $w_i$. In contrast, our approach evaluates only the rank discrepancies between $r_i$ and $w_i$. We define RI as a discrimination property because it captures the fundamental aspect of knowledge-aware refusal. This is because absolute discrepancy-based metrics are sensitive to changes in a model's overall refusal rate, which can significantly affect the metric value (an undesirable property). In contrast, discriminative metrics like RI or AUROC measure only the rank between different samples, making them more robust to changes in overall refusal rates. Alternatively, it is generally easier for a model to adjust its overall refusal rate than to improve its ability to rank questions by difficulty accurately. Next, we introduce how to estimate the Refusal Index through a two-pass evaluation process (Section 3.1).

### 3.1 TWO-PASS EVALUATION

The naive way to measure RI would require the refusal probability $P(f_{\text{LM}}(x_i) = \perp)$ across questions with varying error probabilities. However, in factuality evaluation, we only observe single text output from the model, making refusal probabilities inaccessible. To address this issue, we propose a two-pass evaluation process to infer the Spearman correlation between refusal and error probabilities from binary observations. This approach models refusal decisions by first treating refusal and correctness indicators as results of thresholding on their respective probabilities, and then modeling their joint distribution with a Gaussian copula.

**Formulating the Joint Distribution.** We estimate $\rho_S$ from the joint distribution of refusal and error probabilities using a Gaussian copula model with correlation $\rho$ as follows:

$$C(u, v) = \Phi_\rho\big(\Phi^{-1}(u), \Phi^{-1}(v)\big). \tag{3}$$

Here, $u = F_r(r_i)$ and $v = F_e(w_i)$ are the marginal CDFs, $\Phi^{-1}$ is the standard normal quantile function, and $\Phi_\rho$ is the bivariate standard normal CDF with correlation $\rho$. This Gaussian copula specifies only the dependence on $\rho$ and leaves the marginal distributions of $r_i$ and $w_i$ unrestricted. The function $\Phi_\rho$ depends only on rank correlation, remaining independent of the marginal distributions.

Next, we avoid modeling $F_r$ and $F_e$ directly and instead estimate $\rho$ from $R_i$ and $W_i$ via maximum likelihood. We then compute $\rho_S$ from $\rho$ using the standard conversion formula for Gaussian copulas: $\rho_S = \frac{6}{\pi} \arcsin\left(\frac{\rho}{2}\right)$ (Kendall & Stuart, 1979). Because $\rho$ determines the corresponding rank correlations of $r_i$ and $w_i$ via a monotonic transformation, we could equivalently report other rank measures such as Kendall's $\tau$ instead of $\rho_S$. We use Spearman's $\rho$ for interpretability.

Estimating $\rho$ requires observing two binary indicators for each sample: $R_i$ for refusal probability $r_i$ and $W_i$ for error probability $w_i$, while $r_i$ and $w_i$ themselves remain latent. We achieve this through a two-pass evaluation that runs the model on the same dataset twice. The first pass observes refusal decisions $R_i$, using a standard setup that allows the model to answer or refuse each question, classifying responses as correct, incorrect, or refused. The second pass observes correctness $W_i$ by updating the system prompt to remove abstention options and requiring the model to answer all questions. We provide prompt details in Section B and an illustration in Figure 2. We run the second pass only on questions refused in the first pass, collecting correctness indicators $W_i'$ for all refused questions ($R_i = 1$).

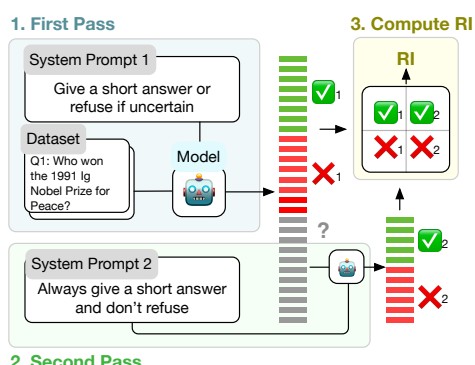

Figure 2: Illustration of two-pass evaluation process.

**Estimating Refusal Index.** We define the aggregated correctness indicator $\hat{W}_i = R_i \cdot W_i' + (1 - R_i) \cdot W_i$ as the correctness when the model provided an answer. The empirical refusal rate is $r = \sum_{i=1}^{|D|} R_i / |D|$ and the error rate is $\mu = \sum_{i=1}^{|D|} \hat{W}_i / |D|$. Under our model, the pair $(R, \hat{W})$ results from thresholding a bivariate standard normal vector $(Z_R, Z_W)$ with correlation $\rho$ at, matching the standard tetrachoric setup implied by the copula.

$$\tau_R = \Phi^{-1}(1 - r), \qquad \tau_W = \Phi^{-1}(1 - \mu). \tag{4}$$

Let $n_{ab}$ be the counts of $(R = a, \hat{W} = b)$ for $a, b \in \{0, 1\}$. The cell probabilities are

$$
\begin{aligned}
p_{11}(\rho) &= \bar{\Phi}_2(\tau_R, \tau_W; \rho) \;=\; P(Z_R > \tau_R, Z_W > \tau_W), \\
p_{10}(\rho) &= r - p_{11}(\rho), \qquad p_{01}(\rho) \;=\; \mu - p_{11}(\rho), \\
p_{00}(\rho) &= 1 - r - \mu + p_{11}(\rho).
\end{aligned}
\tag{5}
$$

We estimate $\hat{\rho}$ by maximizing the multinomial log-likelihood and use $\hat{\rho}$ to compute $\rho_S$:

$$\hat{\rho} = \arg\max_{\rho \in (-1,1)} \ell(\rho), \quad \text{where} \quad \ell(\rho) = \sum_{a,b \in \{0,1\}} n_{ab} \log p_{ab}(\rho). \tag{6}$$

## 4 EXPERIMENTS & RESULTS

### 4.1 EXPERIMENTAL SETUP

**Models.** We evaluate RI on 16 models across different families, sizes, and architectures to ensure comprehensive coverage. Our open-source models include *Gemma-3-12B* (Gemma Team, 2025), *Qwen3-32B/235B* (Qwen Team, 2025) in both think and no-think modes, *Qwen2.5-72B-Instruct* (Qwen Team et al., 2024), *Llama 3.1 70B* (Grattafiori et al., 2024), *Mistral-Large-Instruct-2411* (Mistral AI, 2024), *GLM-4.5 and GLM-4.5-Air*(GLM-4.5 Team et al., 2025) and *DeepSeek-V3-0324* (DeepSeek-AI et al., 2024). Our proprietary models include *Claude 3.5 haiku* (Anthropic, 2024), *Claude Sonnet 4* (Anthropic, 2025), *GPT4.1* and *GPT4.1 mini* (OpenAI, 2025) and *Gemini 2.5 Flash* and *Gemini 2.5 Flash Lite* (Comanici et al., 2025). We use temperature=0.7 and top-p=0.95 across all models. More implementation details are provided in Section C.

**Datasets.** We evaluate RI on three scenarios that require model to make accurate, knowledge-aware refusals: factual question answering, extrinsic hallucination detection (hallucination from training data), and intrinsic hallucination detection (hallucination from context). (1) We use factual question answering to test models' ability to refuse unknown facts. Specifically, we use SimpleQA (Wei et al.,

Table 2: Score variability across different refusal rates. We run evaluation with different refusal tendencies on SimpleQA for each model. $\Delta_{\text{Metric}}$ denotes the normalized difference between most-refusal and least-refusal runs. We use $p = 0.2$ for the Weighted metric. Lower is better.

| Type | Model | $\Delta_{\text{Accuracy}}$ | $\Delta_{\text{Refusal}}$ | $\Delta_{\text{C/A}}$ | $\Delta_{\text{F-score}}$ | $\Delta_{\text{Weighted}}$ | $\Delta_{\text{RI}}$ |
|---|---|---|---|---|---|---|---|
| | Mistral-123B | $-0.40$ | $+0.93$ | $+0.37$ | $-0.16$ | $-0.83$ | $+\mathbf{0.06}$ |
| | Qwen2-35B | $-0.47$ | $+0.95$ | $+0.12$ | $-0.31$ | $-0.62$ | $-\mathbf{0.19}$ |
| Normalized | Qwen2.5-72B | $-0.84$ | $+0.43$ | $+0.50$ | $-0.60$ | $-1.32$ | $-\mathbf{0.07}$ |
| Difference | Qwen3-32B | $-0.96$ | $+0.54$ | $+0.48$ | $-0.71$ | $-1.42$ | $+\mathbf{0.14}$ |
| | Gemma-3-12B | $-1.31$ | $+2.04$ | $+0.96$ | $-0.93$ | $+1.79$ | $+\mathbf{0.42}$ |
| | Average | $-0.80$ | $+0.98$ | $+0.49$ | $-0.54$ | $-0.48$ | $+\mathbf{0.07}$ |
| | **Model** | $CV_{\text{Accuracy}}$ | $CV_{\text{Refusal}}$ | $CV_{\text{C/A}}$ | $CV_{\text{F-score}}$ | $CV_{\text{Weighted}}$ | $CV_{\text{RI}}$ |
| | Mistral-123B | $0.16$ | $0.35$ | $0.14$ | $0.06$ | $0.31$ | $\mathbf{0.04}$ |
| | Qwen2-35B | $0.22$ | $0.47$ | $0.06$ | $0.14$ | $0.32$ | $\mathbf{0.09}$ |
| Coefficient of | Qwen2.5-72B | $0.35$ | $0.17$ | $0.19$ | $0.26$ | $0.53$ | $\mathbf{0.03}$ |
| Variation | Qwen3-32B | $0.35$ | $0.19$ | $0.17$ | $0.28$ | $0.51$ | $\mathbf{0.07}$ |
| | Gemma-3-12B | $0.49$ | $0.76$ | $0.39$ | $0.36$ | $0.66$ | $\mathbf{0.23}$ |
| | Average | $0.31$ | $0.39$ | $0.19$ | $0.22$ | $0.47$ | $\mathbf{0.09}$ |

2024), which contains verifiable, atomic factual questions that challenge even frontier LLMs. (2) We use extrinsic hallucination detection to test whether models correctly refuse to answer when they cannot recall knowledge from training data. For this scenario, we use PreciseWikiQA (Bang et al., 2025), a dynamically generated question-answering dataset from Wikipedia snippets. PreciseWikiQA tests whether models hallucinate information from their training data, assuming Wikipedia knowledge was included during training. We follow Bang et al. (2025) to generate 2000 questions for evaluation. (3) We use intrinsic hallucination detection to test whether models can faithfully recall information with noisy context. For this scenario, we use the 3 datasets from FaithEval (Ming et al., 2025). However, because the `Unanswerable` and `Inconsistency` subsets lack ground truth required for RI computation, we create a 1:1 mixed dataset of PreciseWikiQA and FaithEval to report RI.

**Baseline Metrics.** We compare RI against five established metrics for measuring knowledge-aware refusal (Table 1): Correct Answer Rate, Refusal Rate, Correct given Attempted (C/A), F-score, and Weighted Score. We pick $p = 0.2$ for the Weighted Score to balance the accuracy and refusal rate. We classify all model outputs into three categories following SimpleQA: (1) Correct, (2) Incorrect, or (3) Not Attempted (refusal).

**Adjusting Refusal Rates.** We test RI's consistency by measuring how it changes when models exhibit different refusal rates. To this end, we use different system prompts to instruct models to be more conservative or active in answering questions. These prompts modify refusal tendencies without degrading the quality of refusal decisions, as shown in Section I. Specifically, we use four different system prompts to evaluate each model with varying refusal rates in the first pass, while keep one default prompt that instructs models to answer all questions in the second pass. The complete system prompts are provided in Section B.

## 4.2 Is RI Stable Across Different Refusal Rates?

For each metric M, we summarize stability across the four refusal prompts using two scale-normalized measures:

- **Normalized difference.** $\Delta_{\text{M}} = (\text{M}_{\max} - \text{M}_{\min})/|\text{M}_{\text{mean}}|$, which captures how far the most- and least-refusal runs deviate relative to the average level.
- **Coefficient of variation.** $CV_{\text{M}} = \text{std}(\text{M})/|\text{M}_{\text{mean}}|$, which measures relative dispersion around the mean and allows comparison across metrics with different scales.

In this section, we validate the Refusal Index across different refusal rates to analyze its stability as a metric for knowledge-aware refusal. Our analysis shows two key findings: (1) the Refusal Index conceptualizes and captures intrinsic knowledge-aware refusal ability through an *accuracy-refusal curve*, and (2) the two-pass evaluation returns consistent RI regardless of a model's refusal rate.

**RI Measures Knowledge-aware Refusal with Accuracy-Refusal Curve.** An accuracy-refusal curve quantifies knowledge-aware refusal by plotting correct answer rate against refusal rate for a model

on the same dataset. This trade-off emerges because refusing uncertain questions reduces incorrect answers but simultaneously decreases correct answer numbers due to false refusals. Consequently, models face a trade-off between maintaining correct answers and avoiding incorrect ones. As shown in Figure 3, fixing any metric constant gives a unique iso-score curve in the accuracy–refusal plane, which describes the accuracy-refusal trade-off relationship assumed by the metric.

Iso-RI curves demonstrate two key advantages over heuristic metrics. First, they represent realistic accuracy–refusal trade-offs that match expected model behavior (see Figure 1, left). All iso-RI curves share the same endpoints: maximum correct answers when refusal rate equals zero, and zero correct answers when refusal rate equals one. Correct answer rate continuously decreases as refusal rate increases. Second, RI focuses solely on curve convexity, remaining independent of maximum correct answer rates and refusal rates. This design allows RI to capture how effectively a model preserves correct answers by minimizing false refusals.

For example, when two models have identical maximum correct answer numbers, the model with higher RI will retain more correct answers at any given refusal rate. The mathematical derivation of these properties is provided in Section E. However, heuristic metrics fail to capture this distinction, instead imposing linear accuracy–refusal relationships at fixed scores. Overall, the Refusal Index measures rank calibration in refusal decisions rather than simply rewarding higher accuracy or lower refusal rates, making it distinct from existing metrics. For an empirical view of these curves across multiple models and refusal prompts, Section F visualizes iso-RI contours together with observed accuracy–refusal points.

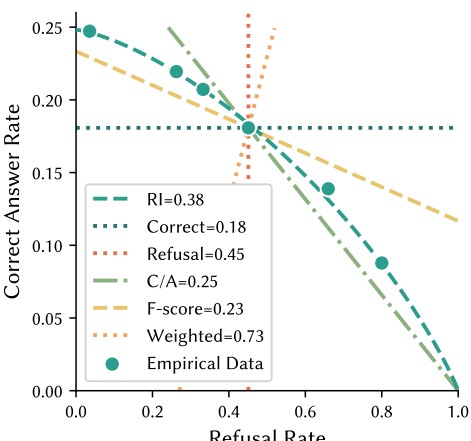

**RI remains consistent across different refusal rates.** We then empirically validate RI by testing its consistency across varying refusal rates. We use 4 system prompts described in Section 4.1 that progressively encourage higher refusal tendency, inducing different refusal rates when applied to the same model on the SimpleQA dataset. Complete results for all models on SimpleQA are provided in Section G. RI

Figure 3: Comparison of factuality metrics with iso-score accuracy-refusal trade-off curves. C/A, F, and W correspond to Correct / Attempted, F-score, and Weighted score, respectively. Empirical data are from Qwen2.5-72B on SimpleQA.

demonstrates high stability across different refusal rates while heuristic metrics show substantial variation. Table 2 shows that RI exhibits approximately 70% lower variability than heuristic metrics. This stability suggests that prompt-induced changes in refusal rate shift the refusal probability distribution without altering the underlying correlation between refusal probability and error probability. We validate this assumption with goodness-of-fit tests in Section D.

## 4.3 IS REFUSAL INDEX FAITHFUL AND CALIBRATED?

**RI is highly consistent with sampling-based calibration methods.** A potential concern arises because RI is defined as rank correlation between refusal probability and error probability, yet the two-pass evaluation may not faithfully capture this correlation. We address this concern by comparing RI values with AUROC scores computed using P(Answering) as an uncertainty estimation method, following Wei et al. (2024). Specifically, we compute P(Answering) by sampling 100 times from each question under temperature=1, then setting the prediction probability to $1 - N_{\text{refusal}}/N$, where $N_{\text{refusal}}/N$ is the ratio of refusal answers in the 100 generations. We then compute AUROC scores between P(Answering) and correctness labels. AUROC with P(Answering) provides a fair comparison because it shares RI's uncertainty definition, measuring only the discriminative ability of refusal as a rank-calibration metric, while P(Answering) directly estimates prediction probability for model refusals. See Appendix L for a complementary reliability-diagram comparison with linear-probe and APRICOT-style calibrators, and for Table 8, which summarizes representative confidence-based methods and RI in terms of bias and computational cost. RI demonstrates the strongest positive correlation with AUROC at 85%, outperforming all other evaluated metrics (Figure 4). This

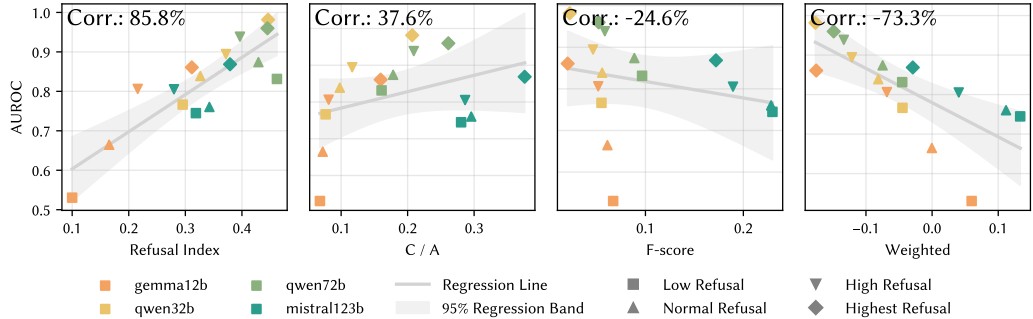

Figure 4: Correlation between factuality metrics and AUROC with P(Answering) on SimpleQA. RI shows the highest positive correlation with AUROC while being much cheaper to compute.

high agreement confirms that RI accurately reflects the correlation between refusal probability and error probability. Additionally, RI requires much lower computational overhead than estimating P(Answering) through multiple sampling.

## 4.4 CAN REFUSAL INDEX CONSISTENTLY RANK MODELS?

We examine whether RI consistently measures knowledge-aware refusal across different models and datasets by analyzing model ranking stability. Ranking stability measures whether RI produces consistent model rankings across different datasets and evaluation settings. Higher ranking stability indicates that a metric captures robust, discriminative model properties. We calculate Kendall's $W$ (overall ranking agreement) and Winner Entropy (top-1 consistency) across 8 evaluation settings: 4 refusal-varying evaluations on SimpleQA plus 4 hallucination benchmarks. Because correct answer rate and refusal rate already provide high ranking stability on their own, we need to filter out their monotonic effects to isolate ranking stability of accuracy-refusal trade-off. Specifically, we perform isotonic regression on correct answer rate or refusal rate across different setups for each model, then remove the regressed values from each metric. These residuals represent metric components that cannot be explained by correct answer rate or refusal rate alone. We then calculate Kendall's $W$ and Winner Entropy on these residuals. We provide detailed procedures in Section K.

**RI provides stable model rankings independent of accuracy and refusal rate.** RI maintains high ranking stability when removing monotonic effects of correct answer rate or refusal rate, while heuristic metrics degrade to near-random stability (Table 3). Heuristic metrics like F-score and Weighted achieve strong ranking stability initially, but their Kendall's $W$ and Winner Entropy drop dramatically after removing monotonic effects from correct answer rate or refusal rate. This pattern reveals that heuristic metrics derive their ranking stability primarily from correct answer rate or refusal rate rather than the relationship between them. However, RI retains most of its ranking stability after removing these effects, demonstrating that it captures intrinsic knowledge-aware refusal properties that persist across different evaluation settings.

Table 3: Ranking stability across different evaluation settings. −Correct and −Refusal show results after removing monotonic effects of correct answer rate and refusal rate with isotonic regression. −Both removes both correctness and refusal rates with additive isotonic regression.

| Metric | Kendall's W ↑ | | | | Winner Entropy ↓ | | | |
| --- | --- | --- | --- | --- | --- | --- | --- | --- |
| | Default | −Correct | −Refusal | −Both | Default | −Correct | −Refusal | −Both |
| Random Value | 0.25 | 0.25 | 0.25 | 0.25 | 0.61 | 0.61 | 0.61 | 0.61 |
| Correct Answer Rate | 0.87 | 0.00 | **0.48** | 0.39 | **0.00** | 1.00 | 0.48 | 1.00 |
| Refusal Rate | 0.86 | 0.44 | 0.00 | 0.30 | 0.18 | 0.61 | 1.00 | 1.00 |
| C / A | 0.69 | **0.63** | 0.40 | 0.37 | 0.33 | 0.61 | 0.48 | 0.61 |
| F-score | **0.90** | 0.10 | 0.48 | 0.39 | 0.00 | **0.18** | 0.48 | 0.52 |
| Weighted | 0.87 | 0.60 | 0.25 | 0.32 | 0.33 | 0.33 | **0.37** | 0.61 |
| RI | 0.47 | 0.50 | 0.35 | **0.49** | 0.47 | 0.33 | 0.61 | **0.37** |

## 5 Discussion

**Does prompting models to be more cautious mitigate miscalibration?** Our results show that prompting strategies have limited impact on knowledge-aware refusal capabilities. LLMs are notorious for overconfidence, answering all questions by default even when they lack knowledge. Instructing models to refuse more questions might seem to help, but our RI analysis reveals otherwise. Table 2 shows that while increasing refusal rates improves the correct answer rate in answered questions (increasing C/A), RI remain stable and far from perfect. This means that, even when a model's refusal rate matches its error rate (eliminating systematic bias), a significant gap persists between actual and perfect refusal decisions. RI quantifies this gap independent of specific refusal rates, providing a stable measure across prompting strategies.

**What factors lead to better knowledge-aware refusal?** We find that model family is the strongest predictor of knowledge-aware refusal ability, surpassing traditional factors like size and accuracy. We found no strong correlation between RI and model parameter sizes, accuracy, or refusal rates within our tested models. Figure 5 plots the relationship between correct answer rate in SimpleQA and average RI scores, with a regression line showing the expected relationship. The correct answer rate shows only $R^2 = 0.235$ correlation with Refusal Index, indicating that higher factual accuracy does not necessarily improve knowledge-aware refusals. Notably, model family strongly

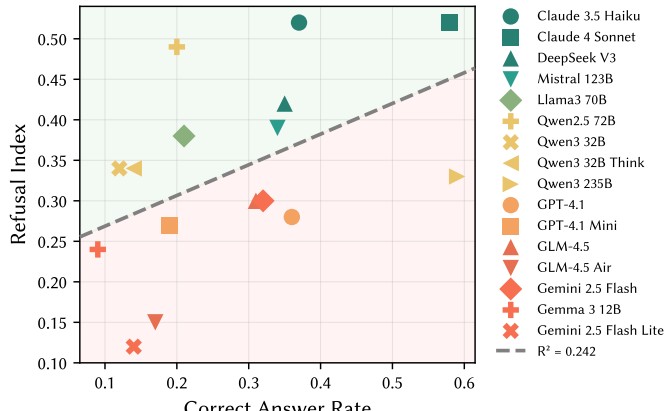

Figure 5: Refusal Index vs. Correct Answer Rate on SimpleQA. The weak correlation ($R^2 = 0.242$) indicates that higher accuracy does not guarantee better knowledge-aware refusal. Models above the regression line (green) outperform expectations; Claude and Qwen models consistently appear in this region, while Gemini, GPT-4.1, and GLM models fall below.

predicts RI performance. Claude and Qwen models (except Qwen 235B) consistently perform above the regression line, demonstrating superior knowledge-aware refusal abilities. In contrast, all Gemini, GPT-4.1, and GLM-4.5 models fall below the regression line. Specifically, Claude models achieve the highest RI scores across both Claude-3.5 Haiku and Claude-4 Sonnet variants. These findings suggest that training pipelines and data distributions used by different model providers play a more critical role in knowledge-aware refusals than model scale or general accuracy.

**Is a model's refusal ability affected by context?** We find that ground truth availability in context significantly impacts knowledge-aware refusal performance, with models struggling most when ground truth is unavailable. We expand RI evaluation to realistic settings where models generate answers conditioned on grounding context with FaithEval. Table 4 presents four scenarios testing different aspects of refusal ability: **PreciseWiki** requires models to recall information from training data; **Counterfactual** tests models' ability to avoid hallucinating from misleading

Table 4: Refusal Index results on hallucination benchmarks.

| Model | Truth Available | | Truth Unavailable | |
| --- | --- | --- | --- | --- |
| | Precise-Wiki | Counter-factual | Incon-sistency | Unans-werable |
| Gemma-3-12b | 0.36 | 0.56 | 0.22 | 0.12 |
| Qwen3-32B | 0.48 | 0.60 | 0.27 | 0.24 |
| Qwen2.5-72B | **0.54** | 0.56 | 0.22 | 0.40 |
| Llama-3.1-70B | 0.52 | **0.70** | 0.17 | 0.31 |
| Mistral-Large | 0.50 | 0.38 | **0.34** | **0.52** |
| Average | 0.48 | 0.56 | 0.24 | 0.32 |

context; **Inconsistency** provides conflicting information requiring refusal; and **Unanswerable** offers no contextual answers. RI values for PreciseWiki are relatively close to those of SimpleQA, and models demonstrate strong ability to identify and avoid counterfactual context. However, when ground truth becomes unavailable (Inconsistency and Unanswerable scenarios), models exhibit substantially worse knowledge-aware refusal. This suggests that knowledge-aware refusal relies on

partial information from training data or context, and degrades when answers never appear in the provided context.

In summary, these findings demonstrate that RI captures an essential dimension of model reliability that is absent from existing factuality metrics. While current factuality evaluation and calibration studies show promising results in improving model accuracy and calibration (Kadavath et al., 2022b), RI reveals a different picture. Our results highlight the need to incorporate knowledge-aware refusal evaluation for comprehensive factuality assessment. We also provide a detailed discussion of the limitations of RI in Section A.

## 6    Related Work

**Factuality evaluation of LLMs.** Factuality evaluation measures an LLM's ability to generate correct answers. Previous methods compare LLM responses against external sources to assess factual correctness (Wei et al., 2024; Min et al., 2023; Kwiatkowski et al., 2019). Many factuality evaluations focus on measuring hallucination, where models generate answers that contradict available information (Bang et al., 2025). Recent work in factuality evaluation recognizes that ground truth may not always be available to the model (Jing et al., 2025). Some works improve factuality by training models to refuse questions beyond their knowledge boundaries (Cao, 2024; Xu et al., 2024; Ouyang et al., 2022). Our metric evaluates calibration through refusal behavior rather than targeting hallucination rate directly.

**Calibration evaluation on black-box models.** Calibration measures the alignment between a model's output probability and its actual probability of being correct (Guo et al., 2017). Calibration serves as a valuable factuality metric because it quantifies a model's self-awareness of its own knowledge (Kadavath et al., 2022a; Yin et al., 2023a; Agrawal et al., 2023). Estimating calibration for black-box LLMs requires inferring uncertainty from text outputs. Previous works propose semantic similarity measures (Kuhn et al., 2023; Farquhar et al., 2024) or auxiliary models (Ulmer et al., 2024) to estimate uncertainty, producing error-based or rank-based calibration metrics (Huang et al., 2024). These methods require training a separate calibrator for each model, making them computationally expensive and model-dependent. Our metric measures the correlation between uncertainty and difficulty, representing a form of rank-based calibration.

## 7    Conclusion

We propose Refusal Index (RI), a novel metric that measures LLMs' knowledge-aware refusal ability through the correlation between refusal decisions and answer incorrectness, addressing critical limitations of existing factuality evaluation methods. Our two-pass evaluation framework provides a practical and lightweight approach to measure RI, enabling more reliable model comparisons independent of accuracy or refusal rate. This work opens new directions for developing better-calibrated AI systems and provides a foundation for evaluating self-knowledge in LLMs.

## Acknowledgements

This work was supported in part by the National Natural Science Foundation of China under Grant 62172123 and Grant 62302122, Heilongjiang Provincial Natural Science Foundation of China under Grant JQ2024F001, and HK RGC under Grants C1043-24GF and RFS2425-1S01.

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

APPENDIX SUMMARY

This appendix provides essential background and technical details supporting our Refusal Index evaluation framework. We first discuss the limitations of our approach in Section A, followed by complete system prompts for the two-pass evaluation methodology in Section B. We then present comprehensive experimental configurations in Section C and compare external calibration methods in Section L. The theoretical foundations are established through validation of the Gaussian copula assumption (Section D) and mathematical derivations of iso-RI curve properties (Section E). Complete experimental results on SimpleQA are provided in Section G. Extended analyses include stability assessments regarding sample size (Section H) and prompt design variations (Section I), along with a ready-to-use Python implementation of our metric (Section J). We conclude with ranking stability evaluation methodology (Section K) and an LLM usage declaration.

## A    LIMITATIONS OF REFUSAL INDEX

The Refusal Index has three key limitations that practitioners should consider. First, the two-pass evaluation requires models capable of following instructions to either refuse questions or provide forced answers, limiting applicability to relatively capable models. Second, our formulation targets knowledge-aware refusal specifically and may not generalize to other refusal types or other applications, such as safety-based refusals, or refusal behavior in non-factual tasks. Finally, knowledge-aware refusal provides a relatively weak signal compared to metrics like correct answer rate, requiring larger datasets for stable RI scores (Section H). Despite these limitations, RI offers a pragmatic metric for an important capability that previous metrics overlooked.

## B    SYSTEM PROMPTS FOR TWO-PASS EVALUATION

We provide the complete system prompts used in our experiments to enable accurate reproduction. These prompts use consistent formatting instructions to standardize outputs. We include in-context learning examples to ensure stable model behavior and syntactically correct answers in the required format.

**Second Pass System Prompt.** The second pass forces models to answer questions that were refused in the first pass. We combine explicit instructions with in-context examples to enforce the output format and minimize formatting errors. Most models rarely refuse when given such instructions, so we simply instruct the model to always provide an answer. The in-context examples help the model consistently produce the required XML-style tags. We show the complete second-pass system prompt in Figure 6.

**First Pass System Prompt.** Our first-pass prompts use the same design and formatting as the second-pass prompt. However, we make small, targeted changes to the instruction wording and the mix of in-context examples to encourage appropriate refusal behavior. Figure 7 shows one complete first-pass prompt variant. Other variants differ only in how strongly they encourage refusal and the proportion of examples that show returning `<answer>UNANSWERED</answer>` when uncertain.

These first-pass variants differ from the second-pass prompt (Figure 6) in only two ways: (1) how strongly the instruction encourages refusal and (2) the proportion of `<answer>UNANSWERED</answer>` responses in the in-context examples. These changes control the overall refusal tendency without otherwise changing the task.

## C    DETAILED EXPERIMENTAL SETUP

This section provides comprehensive details of our experimental methodology to enable reproduction of our results.

**Model inference and generation settings.** To ensure fair comparison across different language models, we maintained consistent decoding hyperparameters throughout our evaluation. Unless explicitly stated otherwise, all models used nucleus sampling with temperature = 0.7, top-p = 0.95, and a maximum generation length of 4096 tokens. We served all open-source models using vLLM

```
Second Pass Prompt

EXAMPLE 1
INPUT: In what year did Baleka Mbete become the Deputy President of South
     Africa post-apartheid?
OUTPUT: <answer>2008</answer>

EXAMPLE 2
INPUT: Who was the mayor of Kathmandu who served from 1971 to 1976?
OUTPUT: <answer>Rajendra Man Suwal</answer>

[omitted for brevity]

EXAMPLE 10
INPUT: The book "Rhine" by Anselm Kiefer is from what year?
OUTPUT: <answer>1981</answer>

Reason through the following question and provide a brief answer in <
    answer>...</answer> at the end. You always give an answer and you
    never return <answer>UNANSWERED</answer>.

Question: {question}
```

Figure 6: Second-pass system prompt used to force answers on previously refused questions. The prompt standardizes output format and includes in-context examples to minimize formatting errors.

```
First Pass Prompt -- Highest Refusal Rate

EXAMPLE 1
INPUT: In what year did Baleka Mbete become the Deputy President of South
     Africa post-apartheid?
OUTPUT: <answer>UNANSWERED</answer>

[omitted for brevity]

EXAMPLE 10
INPUT: The book "Rhine" by Anselm Kiefer is from what year?
OUTPUT: <answer>UNANSWERED</answer>

Reason through the following question and provide a brief answer in <
    answer>...</answer> at the end. You are very cautious and need good
    evidence before drawing conclusions. You prefer saying you don't know
     by returning <answer>UNANSWERED</answer> rather than risking a wrong
     answer.

Question: {question}
```

Figure 7: First-pass prompt variant that induces the highest refusal rate. The instruction and examples bias toward returning UNANSWERED when uncertain.

with eight NVIDIA A800 (80 GB) GPUs, while proprietary models were accessed through their official APIs using identical decoding parameters.

For Qwen3, we evaluated both "thinking" and "non-thinking" modes to assess the impact of chain-of-thought reasoning on knowledge-aware refusal. Qwen3 provides a soft switch mechanism: appending /think or /no_think to the prompt controls the generation of <think> reasoning blocks. Following the official Qwen documentation, we used /no_think to disable thinking mode for non-thinking evaluations.

Table 5: Summary of first-pass prompt variants. Only the refusal instruction and the proportion of UNANSWERED responses differ across variants; all other elements match the second-pass prompt. Ratios vary by model and dataset (see Section G); we report relative levels for brevity.

| Type | Instruction | UNANSWERED ratio |
|---|---|---|
| Low Refusal | You only give an answer if you are confident; otherwise you return <answer>UNANSWERED</answer>. | 0 / 10 |
| Normal Refusal | You are cautious and may return UNANSWERED when unsure. | 1 / 10 |
| High Refusal | You make reasonable guesses from partial information but avoid speculation; return UNANSWERED if not very confident. | 4 / 10 |
| Highest Refusal | You are very cautious and prefer UNANSWERED rather than risking a wrong answer. | 6 / 10 |

**Answer classification and automatic grading.** Our evaluation framework classifies model responses into either correct, incorrect, or refused to enable RI computation. We instruct each model to generate responses with an <answer> tag containing either a factual answer or a refusal with <answer>UNANSWERED</answer>. We first detect refusals by scanning for the exact substring <answer>UNANSWERED</answer>. Remaining responses are classified using the same automatic grading system as SimpleQA to ensure consistency with established benchmarks. We employ the same model used in SimpleQA, GPT-4o-mini with default generation settings provided by OpenAI as our automatic grader, which has demonstrated high reliability in SimpleQA evaluation. The grader classifies each predicted answer as CORRECT, INCORRECT, or NOT_ATTEMPTED using the prompt shown in Figure 8.

This LLM grader handles cases where models make refusals but did not return <answer>UNANSWERED</answer>. In such cases, the grader classifies these responses as NOT_ATTEMPTED based on the content of the predicted answer. In the second pass, we use the same LLM grader but classify NOT_ATTEMPTED responses as INCORRECT, as we do not expect refusals in the second pass. This LLM grader is used for all three evaluation scenarios.

**Benchmark datasets and evaluation scenarios.** Our evaluation encompasses three complementary scenarios that test different aspects of knowledge-aware refusal: factual recall, extrinsic hallucination detection, and intrinsic hallucination detection. This comprehensive approach ensures that RI captures refusal behavior across diverse knowledge-intensive tasks.

**Factual question answering (SimpleQA):** We use SimpleQA to evaluate models' ability to refuse unknown factual information. SimpleQA contains 4,326 carefully curated factoid questions spanning science, geography, history, and popular culture. Each question has a single, indisputable answer verified by two independent annotators with high inter-annotator agreement. This benchmark tests whether models can appropriately refuse questions about facts they may not have learned during training.

**Extrinsic hallucination detection (PreciseWikiQA):** We evaluate models' ability to refuse when they cannot accurately recall information from their training data using PreciseWikiQA from the HalluLens benchmark suite. PreciseWikiQA dynamically generates short factual questions from Wikipedia snippets, assuming that Wikipedia content was included in model training. The evaluation protocol first assesses model refusal decisions, then classifies non-refused answers as correct, incorrect, or unverifiable using an LLM judge. We use the same LLM grader as in SimpleQA for this task.

**Intrinsic hallucination detection (FaithEval):** We assess models' ability to refuse when provided with insufficient or contradictory context using three tasks from FaithEval. This benchmark evaluates knowledge-aware refusal in retrieval-augmented generation scenarios, constructed from ten diverse QA datasets (SQuAD, NewsQA, TriviaQA, NaturalQuestions, SearchQA, HotpotQA, BioASQ, DROP, RACE, and TextbookQA).

The three FaithEval tasks target different contextual challenges:

- **Unanswerable Context:** Context is modified to remove supporting evidence (2.4K examples with >98% human-evaluator agreement on automatic checks)

```
Grade the predicted answer as CORRECT (A), INCORRECT (B), or
    NOT_ATTEMPTED (C).

Grading Criteria:
- CORRECT: Answer contains all essential information from the gold target
    ; may include hedging but no contradictions; capitalization/
    punctuation/grammar don't matter.
- INCORRECT: Answer contradicts the gold target, even with hedging.
- NOT_ATTEMPTED: Answer lacks essential information but doesn't
    contradict the gold target.

Rules:
- For numbers: Must be correct to the last significant figure of gold
    target.
- Information implied by the question doesn't need repeating (e.g., "San
    Francisco" is correct for "What city is OpenAI in?" when gold target
    is "San Francisco, California").
- Minor typos in names are acceptable.
- Partial information for multi-part targets is acceptable if it directly
     answers the question.
- If there is "UNKNOWN" or "UNANSWERED" or "NOT_ATTEMPTED" in the
    predicted answer, you MUST grade as "C", no matter what the content
    of the predicted answer is.

<Begin of Question>
{question}
<End of Question>

<Begin of Gold Target>
{answer}
<End of Gold Target>

<Begin of Predicted Answer>
{predicted_answer}
<End of Predicted Answer>

Return the letter "A", "B", or "C" with no other text. The grade is:
```

Figure 8: LLM grader prompt used to classify predictions as CORRECT, INCORRECT, or NOT_ATTEMPTED for both passes.

- **Inconsistent Context:** Multiple documents with conflicting answers are concatenated (1.5K samples)
- **Counterfactual Context:** Context supports false statements (e.g., "water freezes at 100°C") using multiple-choice questions (1K samples)

Together, these tasks provide 4.9K contextual QA pairs that reveal models' difficulty in maintaining faithfulness to provided context when information is incomplete or contradictory.

## D VALIDATING THE GAUSSIAN COPULA ASSUMPTION

We use the bivariate normal copula (Gaussian copula) to model the joint dependence between refusal and incorrectness under forced answering. This choice allows us to estimate the Refusal Index by capturing the correlation structure between the latent refusal score and question difficulty while remaining agnostic to the marginal distributions. We must validate whether this distributional assumption is appropriate for real model behavior.

We compare the Gaussian copula against three alternative copula families to determine which provides the best fit for modeling refusal behavior. We consider Student-$t$, Gumbel, and Clayton copulas

Table 6: Copula comparison on SimpleQA across model-prompt combinations (ties counted as 0.5). Left panel reports mean goodness-of-fit metrics across all combinations for each copula family. Right panel reports the fraction of combinations where the Gaussian copula outperforms each alternative.

| Goodness-of-Fit Metrics | | | | Gaussian Win Rates | | | |
| --- | --- | --- | --- | --- | --- | --- | --- |
| Family | Log-likelihood | AIC | BIC | Versus | Log-likelihood | AIC | BIC |
| Gaussian | $-1832.08$ | 3666.16 | 3671.76 | Student-$t$ | 1.000 | 1.000 | 1.000 |
| Student-$t$ | $-1859.14$ | 3722.27 | 3733.48 | Clayton | 0.676 | 0.647 | 0.647 |
| Gumbel | $-2086.86$ | 4175.72 | 4181.32 | Gumbel | 0.632 | 0.618 | 0.618 |
| Clayton | $-2200.13$ | 4402.26 | 4407.86 | | | | |

as alternatives, each capturing different forms of dependence structure. We evaluate which copula family best fits the observed refusal patterns across multiple models and prompts.

**Evaluation Criteria.** We use two criteria to evaluate copula performance: goodness-of-fit and win-rate comparisons between the Gaussian copula and the alternatives.

Since the margins are fixed by construction in our two-pass evaluation setup, the natural goodness-of-fit criterion is the multinomial log-likelihood implied by each copula through the resulting $2 \times 2$ cell probabilities. Different copulas have varying numbers of parameters (e.g., Student-$t$ has 2 parameters while Gaussian has only 1), so we must penalize model complexity to ensure fair comparison. We complement the raw log-likelihood with the Akaike Information Criterion (AIC) and Bayesian Information Criterion (BIC):

$$\text{AIC} = 2k - 2\ell(\hat{\theta}), \tag{7}$$

$$\text{BIC} = k \log(n) - 2\ell(\hat{\theta}). \tag{8}$$

where $k$ is the number of parameters, $n$ is the sample size, and $\ell(\hat{\theta})$ is the maximized log-likelihood. These criteria penalize more complex dependence structures, providing a principled basis for model selection.

For the second criterion, we evaluate win rates by comparing how often the Gaussian copula outperforms each alternative across different model-prompt combinations. We compare the Gaussian copula with three standard alternatives that capture different forms of dependence. A Student-$t$ copula adds a heavy-tail parameter to the Gaussian structure; a Clayton copula emphasizes lower-tail association and is asymmetric; and a Gumbel copula emphasizes upper-tail association and is also asymmetric. All candidates are fit by maximum likelihood with margins fixed at the empirical refusal and forced-answering error rates for each model-prompt combination. Win rates are computed across these individual model-prompt units to assess the relative performance of each copula family.

**Experimental Setup.** We systematically evaluate and compare the maximum log-likelihood for each copula family on the SimpleQA dataset. Our evaluation covers all 16 models and 4 first-pass prompts used in the main evaluation (see Section G).

For each model-prompt combination, we obtain a $2 \times 2$ contingency table with margins $(r, \mu)$ representing the refusal rate and error rate respectively. Each copula $C$ maps these margins to cell probabilities $(p_{00}, p_{01}, p_{10}, p_{11})$, and we estimate the copula parameters by maximizing the multinomial likelihood of the observed counts as defined in Equation 9:

$$\hat{\rho} = \underset{\rho \in (-1,1)}{\arg\max} \, \ell(\rho),$$

$$\text{where} \quad \ell(\rho) = \sum_{a,b \in \{0,1\}} n_{ab} \log p_{ab}(\rho). \tag{9}$$

This setup isolates the copula choice while maintaining consistency with the main evaluation framework.

The results in Table 6 show that the Gaussian copula provides the strongest average fit. After accounting for complexity, it provides the best overall trade-off between parsimony and data fit. The Student-$t$ copula, despite its additional heavy-tail parameter, does not improve the average log-likelihood and is uniformly worse once complexity penalties are applied. This aligns with intuition

for $2 \times 2$ data with fixed margins, where heavy tails are weakly identified and tend to degenerate toward the Gaussian case. The asymmetric Clayton and Gumbel copulas trail substantially on both raw fit and information criteria, though they can win occasionally on individual units.

**Conclusion.** We choose the Gaussian copula for two primary reasons: (1) it provides the better average fit across model-prompt combinations as evidenced by superior log-likelihood, AIC, and BIC scores; and (2) it is the simplest and most interpretable copula family, requiring only a single correlation parameter while making minimal distributional assumptions about the dependence structure.

Consequently, the bivariate normal copula is both simple and sufficiently accurate for the refusal–incorrectness dependence considered here. Its combination of low assumptions and competitive fit makes it a natural default for estimating the Refusal Index.

## E    FIXED ENDPOINTS AND SHAPE OF ISO-RI CURVES

We derive two key properties of the accuracy–refusal curve used in the paper: (i) every iso-RI curve passes through the same two endpoints at refusal $r = 0$ and $r = 1$; and (ii) when the association between *wrongness* and the *refusal score* is stronger (i.e., larger RI), the curve is higher in its interior, creating more curvature relative to the straight line joining its endpoints.

Let $(Z_R, Z_W)$ be jointly standard normal with correlation $\rho \in (-1, 1)$. Fix thresholds $\tau_r, \tau_w \in \mathbb{R}$ and define

$$R := \mathbf{1}\{Z_R > \tau_r\} \quad \text{(refuse)}, \qquad W := \mathbf{1}\{Z_W > \tau_w\} \quad \text{(wrong under forced answering).}$$

The refusal rate is $r := \Pr(R = 1) = 1 - \Phi(\tau_r)$. The unconditional error rate is $\pi := \Pr(W = 1) = 1 - \Phi(\tau_w)$, so the correct answer rate (at $r = 0$) is $\mu := 1 - \pi = \Phi(\tau_w)$, where $\Phi$ is the standard normal CDF. For a given $r \in (0, 1)$ we take $\tau_r = \Phi^{-1}(1 - r)$. We define the *correct answer rate* at refusal $r$ as

$$a(r; \rho) := \Pr(\text{correct and answered}) = \Pr(W = 0, R = 0) = \Phi_2\big(\tau_r, \tau_w; \rho\big). \tag{10}$$

where $\Phi_2(\cdot, \cdot; \rho)$ is the bivariate standard normal CDF with correlation $\rho$. We orient the score so that higher $Z_R$ means "more refuse" for items more likely to be wrong (the intended setting for RI, typically $\rho \geq 0$).

**Proposition 1 (Endpoints).** For any $\rho$ and $\tau_w$,

$$a(0; \rho) = \mu \qquad \text{and} \qquad a(1; \rho) = 0.$$

*Proof.* At $r = 0$ we have $\tau_r = +\infty$, hence $a(0; \rho) = \Phi_2(+\infty, \tau_w; \rho) = \Phi(\tau_w) = \mu$. At $r = 1$ we have $\tau_r = -\infty$, hence $a(1; \rho) = \Phi_2(-\infty, \tau_w; \rho) = 0$. $\square$

*Monotonicity in $r$.* Since $\tau_r = \Phi^{-1}(1 - r)$ is strictly decreasing in $r$ and $\Phi_2$ is increasing in each argument, $a(r; \rho)$ is strictly decreasing in $r$ for fixed $\rho$.

This makes intuitive sense: at $r = 0$ we answer everything, so correct answer rate equals the model's overall accuracy $\mu$. As $r \to 1$ we answer almost nothing, so the correct answer rate approaches 0.

**Proposition 2 (Monotonicity in $\rho$).** Fix any interior refusal level $r \in (0, 1)$. Then $a(r; \rho)$ in equation 10 is strictly increasing in $\rho$.

*Proof.* With $r$ fixed, $\tau_r$ is fixed, and $a(r; \rho) = \Phi_2(\tau_r, \tau_w; \rho)$. The standard identity $\frac{\partial}{\partial \rho} \Phi_2(x, y; \rho) = \varphi_2(x, y; \rho) > 0$ implies $\frac{d}{d\rho} a(r; \rho) = \varphi_2(\tau_r, \tau_w; \rho) > 0$. $\square$

**Corollary (Higher curves with higher RI).** All accuracy–refusal curves share endpoints $(r, a) = (0, \mu)$ and $(1, 0)$ by Proposition 1. If $\rho_2 > \rho_1$ (i.e., higher RI), then by Proposition 2, $a(r; \rho_2) > a(r; \rho_1)$ for every $r \in (0, 1)$. Thus the higher-RI curve lies strictly above the lower-RI curve throughout the interior while meeting it at the endpoints, creating greater upward curvature relative to the straight line between $(0, \mu)$ and $(1, 0)$.

The intuition is straightforward: at a fixed refusal level, the key factor in equation 10 is the joint tail probability $P_{11}(\rho)$. As $\rho$ increases, wrong items and high-refusal items occur together more often, making the kept (non-refused) set cleaner. This increases the correct answer rate at every interior $r$. Since the endpoints are fixed, the entire curve shifts upward.

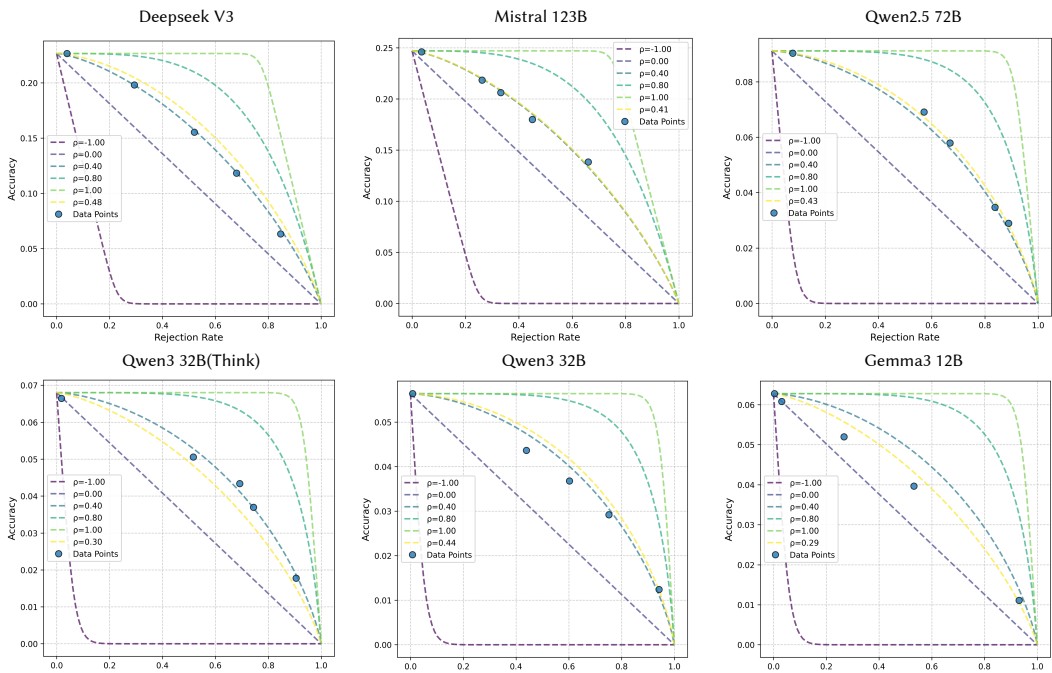

Figure 9: Extended iso-RI visualizations for six models on SimpleQA. Each panel plots empirical accuracy–refusal points from four refusal prompts (dots) together with iso-RI contours (background lines). Models whose points lie close to a single contour have stable Refusal Index across prompts, while widely spread points indicate less consistent refusal behaviour.

## F    EXTENDED ISO-RI VISUALIZATIONS AND FRONTIER MODELS

To complement the theoretical properties above, we provide extended iso-RI visualizations across multiple models. Figure 9 overlays empirical accuracy–refusal points from the four refusal prompts on top of iso-RI contours for six representative models: Gemma-3-12B, Qwen3-32B, Qwen3-32B-Think, Qwen2.5-72B, DeepSeek-V3-0324, and Mistral-123B. For each model, the four points trace out an accuracy–refusal trade-off curve whose curvature matches a single iso-RI contour when RI is stable, and deviates from it when the model's refusal behaviour is less consistent. This visualization makes it easier to see which models preserve correct answers while increasing refusal rates and which ones lose many correct answers due to false refusals.

We also update the frontier-model scatter plot in Figure 5 so that every point is annotated with the corresponding model name. This labeling lets readers directly identify which model families lie above or below the regression line relating RI to correct answer rate, clarifying how training pipelines and architectures influence knowledge-aware refusal.

## G    RESULTS ON SIMPLEQA

We provide metrics on all models on SimpleQA in Table 7, the 95% CI is computed by bootstrap with 1000 samples.

## H    IMPACT OF NUMBER OF QUESTIONS

The estimation of RI is derived from the accuracy and refusal rates of our two-pass evaluation. The stability of RI depends on the number of samples in the evaluation dataset. We assess the stability of RI by measuring its variance across subsets of the evaluation data. We create 50 randomly sampled subsets for various sample sizes (from 50 to 2000) and compute the coefficient of variation (CV) for each size, as shown in Figure 10.

Table 7: Results on SimpleQA with 95% CI.

| Model | Correct Answer Rate | Refusal | C / A | F-score | Weighted | Refusal Index |
|---|---|---|---|---|---|---|
| Gemma-3-12B | 0.05 [0.04, 0.06] | 0.16 [0.14, 0.17] | 0.06 [0.05, 0.07] | 0.06 [0.05, 0.07] | -0.79 [-0.81, -0.77] | 0.25 [-0.07, 0.19] |
| Qwen2.5-72B | 0.05 [0.04, 0.05] | 0.74 [0.73, 0.75] | 0.20 [0.18, 0.22] | 0.07 [0.07, 0.08] | -0.21 [-0.22, -0.20] | 0.49 [0.45, 0.53] |
| Qwen3-32B | 0.03 [0.03, 0.03] | 0.68 [0.67, 0.69] | 0.12 [0.10, 0.15] | 0.05 [0.04, 0.05] | -0.29 [-0.29, -0.28] | 0.34 [0.28, 0.40] |
| Qwen3-32B-Think | 0.04 [0.03, 0.04] | 0.71 [0.70, 0.72] | 0.14 [0.13, 0.16] | 0.06 [0.05, 0.06] | -0.25 [-0.26, -0.24] | 0.34 [0.29, 0.39] |
| Qwen3-235B | 0.38 [0.37, 0.39] | 0.36 [0.35, 0.37] | 0.59 [0.58, 0.61] | 0.45 [0.44, 0.46] | -0.27 [-0.28, -0.26] | 0.33 [0.30, 0.37] |
| Mistral-123B | 0.19 [0.18, 0.19] | 0.43 [0.42, 0.44] | 0.34 [0.32, 0.35] | 0.23 [0.22, 0.24] | -0.39 [-0.40, -0.38] | 0.39 [0.35, 0.42] |
| Llama-3.1-70B | 0.03 [0.02, 0.04] | 0.84 [0.83, 0.86] | 0.21 [0.16, 0.26] | 0.06 [0.04, 0.07] | -0.12 [-0.14, -0.11] | 0.38 [0.28, 0.47] |
| GPT-4.1 | 0.34 [0.32, 0.37] | 0.06 [0.05, 0.07] | 0.36 [0.34, 0.39] | 0.35 [0.33, 0.38] | -0.60 [-0.62, -0.58] | 0.28 [0.19, 0.37] |
| GPT-4.1-mini | 0.13 [0.12, 0.15] | 0.31 [0.29, 0.33] | 0.19 [0.17, 0.21] | 0.16 [0.14, 0.17] | -0.56 [-0.58, -0.54] | 0.27 [0.19, 0.34] |
| Claude-Sonnet-4 | 0.09 [0.07, 0.10] | 0.85 [0.83, 0.86] | 0.58 [0.52, 0.63] | 0.15 [0.13, 0.17] | -0.06 [-0.07, -0.05] | 0.52 [0.45, 0.60] |
| Claude-3.5-Haiku | 0.02 [0.02, 0.03] | 0.93 [0.92, 0.94] | 0.37 [0.29, 0.45] | 0.05 [0.03, 0.06] | -0.04 [-0.05, -0.03] | 0.52 [0.41, 0.63] |
| Gemini-2.5-Flash | 0.19 [0.17, 0.20] | 0.42 [0.39, 0.44] | 0.32 [0.29, 0.35] | 0.24 [0.22, 0.26] | -0.40 [-0.42, -0.37] | 0.30 [0.23, 0.36] |
| Gemini-2.5-Flash-Lite | 0.08 [0.07, 0.09] | 0.41 [0.38, 0.43] | 0.14 [0.12, 0.16] | 0.10 [0.09, 0.12] | -0.51 [-0.53, -0.49] | 0.12 [0.03, 0.20] |
| DeepSeek-V3-0324 | 0.16 [0.14, 0.17] | 0.50 [0.48, 0.53] | 0.32 [0.29, 0.35] | 0.21 [0.19, 0.23] | -0.34 [-0.36, -0.32] | 0.42 [0.36, 0.49] |
| GLM-4.5 | 0.06 [0.05, 0.08] | 0.79 [0.77, 0.81] | 0.31 [0.26, 0.35] | 0.11 [0.09, 0.12] | -0.15 [-0.16, -0.13] | 0.30 [0.22, 0.37] |
| GLM-4.5-Air | 0.05 [0.04, 0.06] | 0.71 [0.69, 0.73] | 0.17 [0.14, 0.20] | 0.08 [0.06, 0.09] | -0.24 [-0.26, -0.23] | 0.15 [0.06, 0.23] |

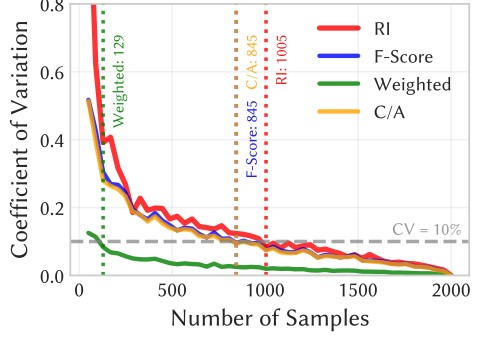

Figure 10: Coefficient of variation of RI when evaluating on subsets of the full dataset.

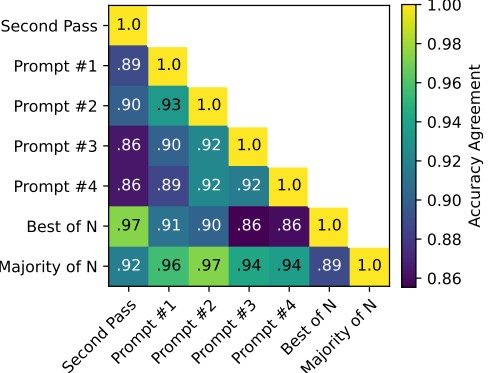

Figure 11: Accuracy agreement between different prompt strategies.

RI is less stable than other metrics with a small number of questions. However, its stability becomes comparable as the sample size increases. To achieve a CV of 0.1, RI requires about 25% more samples than the C/A and F-score metrics. Consequently, a slightly larger number of samples is preferable for obtaining a stable RI estimate.

## I IMPACT OF PROMPT DESIGN

We examine how variations in prompt design affect the RI evaluation. Our experimental setup uses four distinct first-pass prompts, each with different few-shot examples and instructions, to induce varying refusal rates. For the second pass, a single, simpler prompt is used to compel the model to answer all previously refused questions. These prompts are designed to produce different refusal rates. However, we must verify that they do not introduce confounding effects on model accuracy, which would impact the RI calculation.

We measure the accuracy agreement between pairs of prompt strategies to assess this. Agreement is calculated as the proportion of questions for which both prompts yielded the same correctness label, considering only the questions answered by both. The accuracy agreement between different first-pass prompts is consistently high (over 90%), as shown in Figure 11. This indicates that the choice of prompt strategy does not significantly alter the model's underlying accuracy on the questions it chooses to answer. The high agreement involving the forced-answer (second-pass) prompt validates its use for effectively estimating the model's baseline accuracy ($\mu$).

## J   REFUSAL INDEX IMPLEMENTATION

We provide a minimal Python code snippet for computing the Refusal Index (RI) using tetrachoric correlation. This code snippet demonstrates the calculation of RI from two-pass evaluation metrics as described in Section 3, and is shown in Figure 12.

```python
# Refusal Index from two-pass evaluation metrics
from math import log
import numpy as np
from scipy.stats import norm, multivariate_normal
from scipy.optimize import minimize_scalar

def RI(acc1: float, r: float, acc2: float, n: int = 2000) -> float:
    if r <= 0.0 or r >= 1.0:
        return 0.0
    mu = 1.0 - acc2 # wrong rate under forced answering
    acc_att = np.clip(acc1 / max(1e-12, 1.0 - r), 0.0, 1.0)
    mu_a = 1.0 - acc_att # wrong rate on attempted items
    mu_r = float(np.clip((mu - (1.0 - r) * mu_a) / r, 0.0, 1.0)) # wrong
        on refused
    n_r = int(round(n * r)); n_a = n - n_r
    n11 = int(round(n_r * mu_r)); n10 = n_r - n11 # (R=1,W=1),(R=1,W=0)
    n01 = int(round(n_a * mu_a)); n00 = n_a - n01 # (R=0,W=1),(R=0,W=0)
    tau_r, tau_w = norm.ppf(1 - r), norm.ppf(1 - mu)

    def neg_ll(rho: float) -> float:
        rv = multivariate_normal(mean=[0, 0], cov=[[1, rho], [rho, 1]])
        p11 = 1 - norm.cdf(tau_r) - norm.cdf(tau_w) + rv.cdf([tau_r, tau_w
            ])
        p10, p01, p00 = r - p11, mu - p11, 1 - r - mu + p11
        eps = 1e-12
        p11, p10, p01, p00 = [min(1 - eps, max(eps, p)) for p in (p11, p10,
            p01, p00)]
        return -(n11 * log(p11) + n10 * log(p10) + n01 * log(p01) + n00 *
            log(p00))

    rho = minimize_scalar(neg_ll, bounds=(-0.999, 0.999), method="bounded"
        ).x
    return 6 / np.pi * np.arcsin(rho / 2)
```

Figure 12: Minimal Python implementation of the Refusal Index estimator using maximum likelihood to fit the tetrachoric correlation implied by two-pass evaluation statistics.

The function takes three key parameters: `acc1` (accuracy on attempted questions in the first pass), `r` (refusal rate), and `acc2` (accuracy under forced answering in the second pass). The optional parameter `n` represents the total number of questions for statistical estimation. The implementation follows the mathematical framework described in Section 3, using maximum likelihood estimation to find the tetrachoric correlation coefficient that best explains the observed two-pass evaluation results.

## K   RANKING STABILITY METRICS

We use two complementary metrics to evaluate the stability of model rankings across different evaluation settings: Kendall's W and Winner Entropy. These metrics capture different aspects of ranking consistency and are used in Table 3 to assess how reliably different factuality metrics rank models.

### K.1 KENDALL'S W (COEFFICIENT OF CONCORDANCE)

Kendall's W measures the overall agreement among multiple rankings of the same set of items. It quantifies how consistently different evaluation settings (e.g., different refusal rates or benchmarks) rank the models.

Given $m$ evaluation settings ranking $n$ models, let $R_{ij}$ be the rank of model $i$ in evaluation setting $j$. The sum of ranks for model $i$ across all settings is:

$$R_i = \sum_{j=1}^{m} R_{ij}$$

Kendall's W is defined as:

$$W = \frac{12 \sum_{i=1}^{n} (R_i - \bar{R})^2}{m^2 (n^3 - n)}.$$

where $\bar{R} = \frac{m(n+1)}{2}$ is the mean of the $R_i$ values.

Kendall's W ranges from 0 to 1, where:

- $W = 1$ indicates perfect agreement among all rankings
- $W = 0$ indicates no agreement (rankings are essentially random)
- Higher values indicate stronger ranking consistency across evaluation settings

In our evaluation, higher Kendall's W values indicate that a metric produces more stable model rankings regardless of the specific evaluation conditions (e.g., different refusal prompts or datasets).

### K.2 WINNER ENTROPY

Winner Entropy measures the consistency of identifying the top-performing model across different evaluation settings. While Kendall's W considers the entire ranking, Winner Entropy focuses specifically on which model ranks first.

Let $p_i$ be the proportion of evaluation settings where model $i$ ranks first. Winner Entropy is defined as:

$$H_{\text{winner}} = -\sum_{i=1}^{n} p_i \log_n(p_i).$$

where we use base-$n$ logarithm to normalize the entropy to the range $[0, 1]$.

Winner Entropy interpretation:

- $H_{\text{winner}} = 0$ indicates perfect consistency (same model always ranks first)
- $H_{\text{winner}} = 1$ indicates maximum inconsistency (all models equally likely to rank first)
- Lower values indicate more consistent identification of the best model

This metric is particularly important for practical applications where identifying the single best model is the primary concern, rather than the complete ranking.

### K.3 APPLICATION IN OUR ANALYSIS

In Table 3, we apply these metrics to evaluate how different factuality metrics rank models across 8 evaluation settings (4 refusal-varying evaluations on SimpleQA plus 4 hallucination benchmarks). To isolate the ranking stability attributable to accuracy-refusal trade-offs rather than simple accuracy or refusal rate differences, we remove monotonic effects using isotonic regression before computing these metrics. This ensures we measure genuine stability in how metrics capture knowledge-aware refusal rather than stability derived from consistent accuracy or refusal patterns.

### K.4 ISOTONIC REGRESSION PROCEDURE

To isolate the components of factuality metrics that cannot be explained by correct answer rate or refusal rate alone, we employ isotonic regression to remove monotonic effects from these baseline metrics. This procedure allows us to focus on how well each metric captures the intrinsic accuracy-refusal trade-off relationship.

**Individual Metric Regression** For each model $i$ and factuality metric $M$, we have metric values $M_i^{(1)}, M_i^{(2)}, \ldots, M_i^{(k)}$ across $k$ evaluation settings. Similarly, we have corresponding correct answer rates $C_i^{(1)}, C_i^{(2)}, \ldots, C_i^{(k)}$ and refusal rates $R_i^{(1)}, R_i^{(2)}, \ldots, R_i^{(k)}$ for the same model across these settings.

To remove the monotonic effect of correct answer rate, we perform isotonic regression to find the isotonic function $f_C$ that minimizes:

$$\sum_{j=1}^{k} (M_i^{(j)} - f_C(C_i^{(j)}))^2$$

subject to the constraint that $f_C$ is non-decreasing (or non-increasing, depending on the expected monotonic relationship). The residual metric values after removing correct answer rate effects are:

$$M_i^{(j),-C} = M_i^{(j)} - f_C(C_i^{(j)})$$

Similarly, to remove refusal rate effects, we find isotonic function $f_R$ and compute:

$$M_i^{(j),-R} = M_i^{(j)} - f_R(R_i^{(j)})$$

**Additive Isotonic Regression** To remove both correct answer rate and refusal rate effects simultaneously, we employ additive isotonic regression. This approach models the metric as the sum of monotonic functions of both variables plus a residual term:

$$M_i^{(j)} = g_C(C_i^{(j)}) + g_R(R_i^{(j)}) + \epsilon_i^{(j)}$$

We find isotonic functions $g_C$ and $g_R$ that minimize:

$$\sum_{j=1}^{k} (M_i^{(j)} - g_C(C_i^{(j)}) - g_R(R_i^{(j)}))^2$$

subject to monotonicity constraints on both $g_C$ and $g_R$. This optimization is performed using coordinate descent, alternately optimizing $g_C$ while holding $g_R$ fixed, and vice versa, until convergence.

The residual metric values after removing both effects are:

$$M_i^{(j),-\text{Both}} = M_i^{(j)} - g_C(C_i^{(j)}) - g_R(R_i^{(j)})$$

These residuals represent the portion of each metric that cannot be explained by monotonic relationships with correct answer rate or refusal rate, allowing us to assess the intrinsic stability of how each metric captures knowledge-aware refusal properties. The ranking stability metrics (Kendall's W and Winner Entropy) are then computed on these residuals across all models and evaluation settings.

## L COMPARISON WITH EXTERNAL CALIBRATION METHODS

Section 2 argues that external confidence calibrators—such as linear probes, auxiliary models, or sampling-based confidence—do not necessarily reflect the refusal decisions that a model actually makes. Here we provide an ablation on Qwen3-32B to compare three representative confidence estimators on the same mixed factual QA set and relate them to the Refusal Index (RI).

Table 8: Comparison of confidence-based calibration methods and Refusal Index (RI). $N$ is the number of evaluation questions, $S$ is the number of samples per question, and $N_{\text{train}}$ is the number of training samples for auxiliary estimators.

| Method | Typical calibration method(s) | Unbiased | Computational cost |
|---|---|---|---|
| Linear Probe | Train a linear classifier on hidden states | × | $SN_{\text{train}}d$ probe training + $N$ generations + $N$ inferences |
| Black-box Estimator | Auxiliary classifier on output text | × | Calibrator training on $N_{\text{train}}$ samples + $N$ generations + $N$ inferences |
| Verbalized Confidence | Ask model to output numeric confidence | × | $N$ generations + $N$ confidence-score generations |
| Sampling-based | Use refusal frequency to approximate refusal probability | ✓ | $SN$ generations |
| Refusal Index (ours) | Two-pass evaluation, no auxiliary model | ✓ | $2N$ generations |

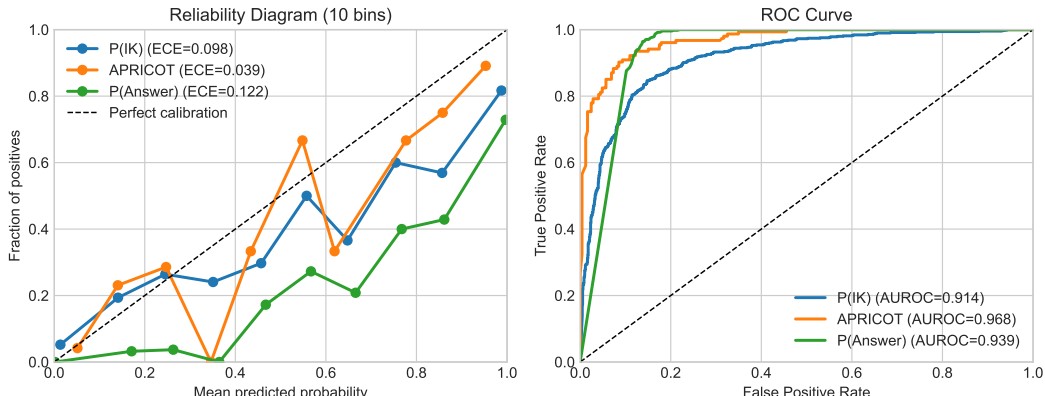

Figure 13: **Calibration comparison on Qwen3-32B.** Reliability diagrams (left, 10 bins, lower ECE is better) and ROC curves (right, higher AUROC is better) for three confidence estimation methods: a white-box linear probe $P(\text{IK})$ (Kadavath et al., 2022a), APRICOT (Ulmer et al., 2024), and sampling-based $P(\text{Answering})$ (Wei et al., 2024).

**Experimental setup.** We compare three representative calibration approaches. $P(\text{IK})$ represents white-box methods, using a linear classifier trained on the model's internal hidden states to predict whether it knows the answer (Kadavath et al., 2022a). **APRICOT** represents auxiliary model-based methods, estimating confidence by analyzing the model's generated reasoning traces with a fine-tuned external model (Ulmer et al., 2024). $P(\text{Answering})$ represents sampling-based methods, estimating confidence by measuring how frequently the model chooses to answer versus refuse across multiple samples for the same question (Wei et al., 2024). We evaluate all methods on the same held-out questions to compare their calibration performance.

**Observations.** Figure 13 shows that the three estimators agree on ranking (ROC curves with AUROC $> 0.91$), but disagree strongly on calibration shape. The linear probe and APRICOT both appear almost perfectly calibrated (ECE 0.098 and 0.039) and would suggest that Qwen3-32B is very well calibrated. In contrast, $P(\text{Answering})$ exhibits a noticeably higher ECE (0.122) and a reliability curve that drops below the diagonal at high predicted probabilities, revealing a clear over-confidence bias in the high-confidence regime. This diagnosis matches the moderate RI of Qwen3-32B on SimpleQA (RI $\approx 0.34$; see Table 7), which indicates substantial room for improving knowledge-aware refusal.

Taken together with prior work showing that verbalized confidence, sampling-based confidence, and auxiliary calibrators can give inconsistent answers about the same model (Wei et al., 2024; Huang

et al., 2024), these results highlight two points: (1) different external calibrators can hide or expose over-confidence depending on how they are constructed, and (2) the sampling-based method that directly uses refusal frequency is the only one whose calibration profile aligns with RI, but it is substantially more expensive to compute. RI therefore provides a cheaper, calibrator-free way to capture the same over-confidence behaviour, using only two standard evaluation passes without additional probes, auxiliary models, or heavy sampling.

## M    LLMs Usage Statement

During the preparation of this paper, we used LLMs (e.g., ChatGPT) for limited assistance with: (1) proofreading and suggesting edits for grammar issues; (2) formatting LaTeX tables from raw data; (3) generating boilerplate code for dataset loading, logging, and plotting; and (4) identifying relevant prior work during literature review. **LLMs were not used for generating paper content, developing ideas or experimental designs, or implementing core evaluation code beyond standard auto-completion. All research contributions, experimental results, and written content are the authors' original work.**

