# OpenReview forum: "Can LLMs Refuse Questions They Do Not Know? Measuring Knowledge-Aware Refusal in Factual Tasks"
_ICLR.cc/2026/Conference — ICLR 2026 Poster_

### Official Review · Reviewer_5y9a · 2025-10-25

**Soundness:** 2
**Presentation:** 1
**Contribution:** 2
**Rating:** 2
**Confidence:** 3

**Summary:**

This paper makes a contribution to the field of LLM evaluation, specifically addressing the critical issue of "knowledge-aware refusal."  Firstly, the authors clearly identify a major gap in existing evaluation methodologies for LLM refusal behavior. Therefore, they propose the primary contribution-the Refusal Index (RI), a rigorously defined metric based on Spearman’s rank correlation between a model's refusal probability and its error probability. This metric is designed to directly and faithfully quantify a model's intrinsic capability to refuse questions beyond its knowledge. The extensive empirical validation across 16 models and 5 datasets demonstrate that RI is stable, consistent, and independent of a model's overall accuracy and refusal rate.

**Strengths:**

1. The paper tackles a critical and under-explored problem in LLM reliability—"knowledge-aware refusal." The introduction of the Refusal Index (RI) is a novel and timely contribution that addresses a clear gap in the existing evaluation landscape.

2. The paper provides extensive and compelling experimental evidence to support its claims. Experiments across 16 models and 5 datasets offer a robust demonstration of RI's stability, consistency, and superiority over existing metrics, and further delivers some insightful ideas.

3. The proposed two-pass evaluation method for estimating RI is lightweight and practical. This thoughtful design makes the metric feasible for researchers and practitioners to adopt without requiring excessive computational resources, enhancing its potential impact.

**Weaknesses:**

1. Although the empirical results of this paper is promising, the technical contribution seems not solid and sound. The rationality of the method is not well presented. Therefore, the technical validity is not convincing.

2. The paper is not well written. There are many concepts introduced in this paper. However, these concepts are not rigorously clarified. The details can be found in Questions.

**Questions:**

1. In section 2.1, why choose bivariate gaussian distribution to model the probablilty value of error and refusal. Does some empirical results or previous works support this point? I think it is a rough characterization.

2. How does the rank in the definition of spearman's rank correlation play the role in the estimation of RI?

3. In equation (3), how can we obtain the value of $r_i$ and $w_i$ ?

---

> ### Author Response · Authors · 2025-11-21
>
> We thank the reviewer for the valuable feedback. We are encouraged that you acknowledge the importance of the problem we tackle and the extensiveness of our experiments.
>
> We understand your concerns regarding the technical rationale and clarity of our method. We would like to carefully clarify the mathematical design choices and demonstrate the soundness of RI.
>
> ---
>
> > **Q1:** In section 2.1, why choose bivariate gaussian distribution to model the probability value of error and refusal?
>
> Thank you for raising this important question. We believe there may be a slight misunderstanding here: **we do not model the probability values of error and refusal using a Bivariate Gaussian Distribution.** Instead, we use a **Gaussian Copula** (Section 3.1). We would like to clarify why this distinction matters and why we chose the Gaussian copula to measure RI:
>
> - **The Challenge:** Ideally, to measure knowledge-aware refusal, we need the correlation between the latent refusal probability $r_i$ and error probability $w_i$. However, in standard evaluations, we only observe binary outcomes (Refused/Answered, Correct/Incorrect), making $r_i$ and $w_i$ unobservable without expensive and noisy calibration steps.
> - **Our Solution (Copula):** The key question is: if we only have binary outcomes for refusal ($R_i$) and error ($W_i$), how can we measure their correlation? This is where the Gaussian Copula becomes essential. It allows us to model the **dependence structure** between two variables independently of their marginal distributions. Crucially, by using a Gaussian Copula, we do **not** assume that $r_i$ and $w_i$ themselves follow a Gaussian distribution. Rather, we use the Gaussian Copula framework to solve for the correlation parameter $\rho$. We described this algorithm in Section 3.1 of the manuscript.
> - **Theoretical Soundness:** This approach is mathematically equivalent to the **Tetrachoric Correlation Model**, a well-established statistical method for estimating the correlation between two latent continuous variables given only binary data [1].
> - **Empirical Validation:** In **Appendix D (Table 6)**, we empirically validated this choice by comparing the Gaussian Copula against Student-t, Clayton, and Gumbel copulas. The Gaussian Copula achieved the best goodness-of-fit (Log-likelihood and AIC/BIC) across our datasets.
>
> We agree with you that a bivariate Gaussian distribution would be too restrictive, and that is precisely why we use a Copula framework instead of assuming a joint distribution. **Following your feedback, we have revised the manuscript to more explicitly clarify this distinction and prevent future confusion. [[🔗L211-L214]](https://openreview.net/pdf?id=9gJBhkLRat#nameddest=5y9a-Q1)**
>
> ---
>
> > **Q2:** How does the rank in the definition of Spearman's rank correlation play a role in the estimation of RI?
>
> This is an excellent question. The "rank" aspect is intrinsic to our use of the Copula framework.
>
> We use Copula to model the dependence between the CDFs of the variables rather than the variables themselves. The parameter $\rho$ derived from the Gaussian Copula corresponds directly to the rank correlation between the underlying latent variables ($r_i$ and $w_i$). This aligns with our definition of RI as Spearman's $\rho_S$. In Definition 3.1, we discuss the rank property of RI. To make this connection more explicit, **we have improved the revisions to further clarify the link between $\rho$ and rank correlation [[🔗L216-L218]](https://openreview.net/pdf?id=9gJBhkLRat#nameddest=5y9a-Q2)**.
>
> ---

---

> ### Author Response · Authors · 2025-11-21
>
> > **Q3:** In equation (3), how can we obtain the value of $r_i$ and $w_i$?
>
> We apologize if the notation caused confusion. The key point is: **we do not calculate individual $r_i$ and $w_i$ values for every sample.**, which can be seen from [[L214]](https://openreview.net/pdf?id=9gJBhkLRat#nameddest=5y9a-Q1): _we avoid modeling $F_r$ and $F_e$ directly and instead estimate $\rho$ from $R_i$ and $W_i$._
>
> Estimating $r_i$ and $w_i$ for every single question would require expensive sampling and would introduce significant noise—this is exactly the problem RI aims to solve. Instead, our method uses a "two-pass" estimation:
>
> 1. We collect the **aggregate** binary outcomes ($R_i$ and $W_i$) across the dataset.
> 2. We use **Maximum Likelihood Estimation (MLE)** on these aggregate counts (Eq. 4-6) to directly estimate the correlation $\rho$ that best explains the observed binary data under the Gaussian Copula framework.
>
> **In the revision, we have updated the explanation to clarify that we bypass the need for individual $r_i$ and $w_i$ estimation [[🔗L221-L223]](https://openreview.net/pdf?id=9gJBhkLRat#nameddest=5y9a-Q3)**.
>
> ---
>
> > **W2:** The paper is not well written. concepts are not rigorously clarified.
>
> We appreciate your feedback, and we sincerely hope the above clarifications and revisions make our technical approach more intuitive and sound. We have worked to improve the clarity throughout the manuscript, particularly in sections describing the copula framework and the relationship between binary outcomes and correlation estimation. Please let us know if you have any remaining questions or concerns. We are happy to provide further clarification!
>
> [1] Olsson, U. Maximum likelihood estimation of the polychoric correlation coefficient. _Psychometrika_ 44, 443–460 (1979).

---

> > ### Author Response · Authors · 2025-11-26
> > **Gentle Reminder Regarding Our Rebuttal**
> >
> > We would like to kindly follow up regarding our rebuttal. We deeply appreciate your careful review, which has helped us strengthen our submission. We understand this is a busy period for reviewers and are grateful for the time you have dedicated to our paper. Please do not hesitate to raise any remaining concerns.

---

### Official Review · Reviewer_y9ka · 2025-10-28

**Soundness:** 3
**Presentation:** 3
**Contribution:** 2
**Rating:** 4
**Confidence:** 4

**Summary:**

This paper aims to evaluate a model’s level of refusal solely based on its output text. To this end, the authors propose a new evaluation metric called the Refusal Index (RI). Unlike traditional metrics that are heavily affected by the refusal rate, RI remains stable across different refusal rates. The paper validates the effectiveness of RI on multiple models and datasets.

**Strengths:**

1.This paper observes that traditional metrics are highly affected by the refusal rate and proposes a new metric called RI.

2.The overall writing of the paper is clear and fluent.

3.The experiments are thorough, and the effectiveness of RI is validated across multiple models and datasets.

**Weaknesses:**

1. The paper lacks sufficient explanation of traditional evaluation metrics. In the introduction, the authors briefly mention some of their weaknesses, but it remains unclear what these metrics actually are and why they exhibit such shortcomings. This background is essential for understanding the motivation of your proposed approach. I suggest moving this part to Section 2 and clearly introducing the limitations of traditional metrics before presenting your own.

2. The authors claim that external calibrators, such as verbalized confidence or linear probes, cannot replace direct refusal measurements. However, the rationale for this statement is not clearly articulated. Could the authors elaborate on why these methods are unsuitable? For instance, since we can explicitly train models to output verbalized confidence, it is not immediately clear why such signals cannot serve as a proxy for model confidence or be used in place of direct refusal measures.

3. Starting around line 129, the definition of the key notion raises potential confusion: are refusals also counted as errors? This point needs clearer explanation. Later sections suggest that each question has its own error rate (and refusals are re-answered for measurement), but this is not obvious when first introduced. Additionally, in Table 1, the formula c/(1–r) is unclear — does c represent the number of correct answers among the non-refused cases? Please clarify this.

4. The proposed metric seems conceptually related to AUROC, which also reflects the consistency between confidence and ability. This is similar to your statement that “its refusal probability increases monotonically with error probability.” Could the authors explain more explicitly how their metric fundamentally differs from AUROC? Why should I use RI instead of AUROC? Refusal probability can also be reflected through confidence scores, rather than being binarized into a simple “refuse or not” decision.

5. The “refusal tendency” appears analogous to a fine-grained confidence estimation, while the “error tendency” seems related to accuracy over multiple responses. Have the authors explored this connection? Why not estimate model confidence directly and then threshold it according to user preferences, instead of measuring refusal explicitly.

6. Since the proposed metric relies on Gaussian estimation, I am concerned about its robustness under limited sample sizes. How accurate is the estimation in such cases? Furthermore, is the Gaussian assumption itself empirically justified?

7. The choice of baselines could be further discussed. For fine-grained confidence, AUROC and ECE are typically appropriate. For binary confidence, accuracy (as a measure of ability) and alignment (whether refusal matches correctness) might be more relevant.
Clarifying why your chosen baselines are suitable would strengthen the experimental section.

8. While the paper discusses the influence of refusal rate, accuracy also substantially affects metric behavior. For example, AUROC can appear artificially high under extremely imbalanced accuracy (e.g., only 10 correct samples out of 1000). This suggests that model ability significantly impacts evaluation. I am also curious about the authors’ perspective on how model competence affects alignment — if a model is either very strong or very weak, learning when to refuse may become trivially easy. Is this an expected or desirable property?

**Questions:**

See Weaknesses.

---

> ### Author Response · Authors · 2025-11-21
>
> We thank the reviewer for their thoughtful feedback and constructive suggestions. We carefully address each concern below.
>
> ---
>
> > **W1:** The paper lacks sufficient explanation of traditional evaluation metrics.
>
> Thank you for raising this point. We agree that this background is essential for understanding our contribution. In the original submission, traditional metrics were mentioned in the Introduction and listed in Table 1. In the revision, **we have added a dedicated "Background" section after the Introduction [[🔗L093-L127]](https://openreview.net/pdf?id=9gJBhkLRat#nameddest=y9ka-W1)**, where we (1) introduce traditional metrics in more detail, relocating Table 1 to this section; and (2) explain why these metrics are insufficient for measuring knowledge‑aware refusal. In summary:
>
> - **Refusal Rate:** This metric captures only the **frequency** of refusals, not the **quality** of those decisions (i.e. whether refusals correlate with errors). Moreover, a model's refusal rate can be trivially manipulated through prompts encouraging cautious behavior.
> - **Heuristic Combinations (e.g., F-score):** These metrics combine refusal and correctness in ways primarily designed to penalize over-refusal. We show that such combinations, including the SimpleQA F‑score [3], can vary by up to ~70% when only the refusal tendency is changed, demonstrating that they conflate accuracy with refusal rate rather than measuring the alignment between them.
>
> ---
>
> > **W2 & W5:** Why not estimate confidence and threshold it (e.g., via AUROC) instead of measuring refusal explicitly?
>
> Thank you for this insightful question, which touches on a core aspect of our motivation. We understand your concern to be: why not use confidence scores and threshold them to approximate refusal behavior?
>
> The fundamental distinction is that **calibration** and **knowledge‑aware refusal** are related but different:
> - Calibration methods produce numeric probabilities of correctness and evaluate how well those probabilities match actual accuracy.
> - Knowledge‑aware refusal (RI) evaluates how well the model's **actual refusal decisions** align with the probability of being incorrect.
>
> **The challenge with external calibrators:** External calibrators cannot replace direct refusal measurements because they often **do not reflect the actual refusal decisions made by models**. Even if we instruct or train models to output confidence scores, thresholding those scores does not necessarily approximate the model's real refusal behavior.
>
> **We support this claim in two ways in the revised paper [[🔗L128-L151]](https://openreview.net/pdf?id=9gJBhkLRat#nameddest=y9ka-W2/W5)**: (1) evidence from prior research, and (2) a new ablation experiment.
>
> - **Prior work shows strong disagreement among calibrators**: SimpleQA [3] finds that verbalized confidence is severely over‑confident, while sampling‑based $P(\text{Answering})$ is moderately well‑calibrated; while APRICOT [2] can appear nearly perfectly calibrated. [5] points out that it becomes unclear which results to trust, as they rely on different assumptions of calibrators and differ greatly in scores. **We also added a new table in the revision, comparing RI with other calibration methods. [[🔗L1296-L1312]](https://openreview.net/pdf?id=9gJBhkLRat#nameddest=hVoC-W2/Q3)** Please see the table below:
>
> 	| Method                   | Typical calibration method(s)                       | Unbiased to true refusal probability? | Computational cost                                                    |
> 	| ------------------------ | --------------------------------------------------- | ------------------------------------- | --------------------------------------------------------------------- |
> 	| Linear Probe             | Train a linear classifier on internal hidden states | ✗                                     | $S*N_\text{train}*d$ Probe training + $N$ generations + $N$ inferences                 |
> 	| Black-box Estimator      | Train an auxiliary classifier on output text        | ✗                                     | Calibrator training on $N_\text{train}$ samples + $N$ generations + $N$ inferences |
> 	| Verbalized Confidence    | Ask model to output a numeric confidence value      | ✗                                     | $N$ generations + $N$ confidence-score generations                        |
> 	| Sampling-based           | Use refusal frequency to approximate refusal prob.  | ✓                                     | $S*N$ generations (S: samples per question)                         |
> 	| **Refusal Index (ours)** | Two-pass evaluation, no extra model                 | ✓                                     | $2N$ generations                                                    |

---

> ### Author Response · Authors · 2025-11-21
>
> - **New experiment**: To empirically validate this point, we added an ablation study using Qwen3‑32B, comparing three estimators:
> 	- $P(IK)$: White-box linear probe confidence estimator [1]
> 	- `APRICOT`: Accurate black-box confidence estimator [2]
> 	- $P(\text{Answering})$: Sampling-based refusal frequency method [3], which makes the fewest assumptions about calibration and has been shown to align with RI
>
> 	**[[🌐 View comparison of calibration methods]](https://cdn-uploads.huggingface.co/production/uploads/62cd3a3691d27e60db0698b0/vzzdjNXM2nF3Kxh3gnldi.png)**
>
> 	The results show that their calibration curves and ECE values differ substantially. Only $P(\text{Answering})$ aligns well with RI, yet RI reaches similar conclusions using just two evaluation passes instead of 100 samples per question. We have incorporated these findings into the Background section of the revision.
>
> ---
>
> > **W3:** Clarification on error definitions and Table 1 formulas. Are refusals also counted as errors? The formula c/(1–r) is unclear — does c represent the number of correct answers among the non-refused cases?
>
> Thank you for identifying this ambiguity. We clarify as follows:
>
> - **Refusals as errors:** Yes, refusals are counted as "incorrect" in our formulation. As stated in Section 3.1: "we define two indicators: $W_i = \mathbf{1}\{f_{\text{LM}}(x_i) \neq y_i\}$ for incorrect outputs and $R_i = \mathbf{1}\{f_{\text{LM}}(x_i) = \bot\}$ for refusal responses." Since $\bot \neq y_i$, all refusals are counted as incorrect. We have made this more explicit in the revised text.
> - **Re-answering:** The re-answering process, which checks whether a refusal was warranted, is captured by $W'_i$, as described in the methodology.
> - **Table 1 notation:** Regarding the formula $c/(1-r)$, note that **$c$ represents the global correct answer rate** (the proportion of correct answers among _all_ samples), as indicated in the first row. This differs from $C/A$, which measures accuracy only among answered questions. **We have revised the Table 1 caption to clarify this distinction [[🔗L108-L118]](https://openreview.net/pdf?id=9gJBhkLRat#nameddest=y9ka-W3)**.
>
> ---
>
> > **W4**: RI seems closely related to AUROC. How does it differ, and why use RI instead?
>
> We agree that RI shares conceptual similarities with AUROC—both are rank‑based discrimination metrics. RI should be understood as a **refusal‑specific, sample‑efficient analogue** of AUROC.
>
> The key differences are:
> - AUROC requires a **real‑valued confidence score**, which for black‑box LLMs must be constructed using a chosen calibrator (e.g., verbalized confidence, linear probe, sampling, APRICOT). The resulting AUROC therefore evaluates the quality of that particular score.
> - RI is defined as the **Spearman correlation between latent refusal probability and error probability**, estimated **directly from binary refusal decisions and correctness labels** via a Gaussian copula, without requiring any auxiliary calibrator.
>
> Empirically, RI is highly correlated (≈85%) with AUROC computed from the sampling‑based $P(\text{Answering})$ method [3], yet requires only two evaluation passes rather than 100 samples per question (Figure 4). Thus, RI captures essentially the same discriminative structure as a strong sampling‑based calibration approach while being far more practical to deploy [3,5].

---

> ### Author Response · Authors · 2025-11-21
>
> > **W6:** Robustness of the Gaussian estimation under limited sample sizes.
>
> Thank you for raising this methodological concern.
>
> - **Gaussian copula vs. Gaussian distribution:** We use a Gaussian _copula_ to model the dependency structure (correlation) between refusal and error. Critically, this does _not_ assume that the marginal distributions of refusal or error are Gaussian—it only models their correlation structure. This approach is standard practice for estimating tetrachoric correlation [4].
> - **Goodness-of-fit:** To empirically validate this choice, we have provided an **ablation experiment in Appendix G** comparing the Gaussian copula against other copula families (e.g., Clayton, Student-t). The results demonstrate that the Gaussian copula consistently provides stable and superior fit for this task. We have clarified this distinction and added the relevant citation in the revision.
>
> ---
>
> > **Q7:** The choice of baselines could be further discussed.
>
> Thank you for this question.
>
> - **Binary confidence metrics:** The metrics in Table 1 (Accuracy, C/A, F-score) represent the standard baselines currently used in factual QA evaluation, particularly in SimpleQA [3].
> - **Fine-grained confidence metrics:** We compare against AUROC computed with $P(\text{Answering})$ because it is the most robust rank-based metric available for this task [3]. We excluded ECE because it measures absolute calibration error rather than the discriminative ranking ability that RI is designed to assess.
>
> **In the revision, we have relocated Table 1 to the Background section and expanded the discussion to explicitly justify these baseline choices. [[🔗L108-L118]](https://openreview.net/pdf?id=9gJBhkLRat#nameddest=y9ka-Q7)**
>
> ---
>
> > **Q8:** Impact of accuracy/model competence and authors' perspective on how model competence affects alignment.
>
> This is an excellent question. One might expect that highly capable models would trivially "solve" refusal, but our findings suggest otherwise.
>
> - **Independence from accuracy:** As shown in **Table 3**, RI maintains high ranking stability even after we mathematically remove the effects of accuracy using isotonic regression. In contrast, heuristic metrics like F-score degrade to near-random performance when accuracy effects are controlled for. This demonstrates that RI measures the _quality of the refusal mechanism itself_, and **this property cannot be explained with merely the model's overall competence**.
> - **Family vs. capability:** **Figure 5** exactly plots competence (accuracy) against alignment (RI), showing that alignment does not scale linearly with accuracy. Some highly accurate model families still exhibit poor refusal alignment. This supports a key insight of our work: knowledge-aware refusal appears to result from specific training methodologies rather than being an automatic byproduct of general capability.
>
> We hope these clarifications and the additional experimental evidence fully address your concerns. Please let us know if you have any further questions!
>
> **References:**
>
> [1] Kadavath, Saurav, et al. "Language models (mostly) know what they know." _arXiv preprint arXiv:2207.05221_ (2022).
>
> [2] Ulmer, Dennis, et al. "Calibrating Large Language Models Using Their Generations Only." _ACL_ (2024).
>
> [3] Wei, Jason, et al. "Measuring short-form factuality in large language models." _arXiv preprint arXiv:2411.04368_ (2024).
>
> [4] Olsson, U. "Maximum likelihood estimation of the polychoric correlation coefficient." _Psychometrika_ 44, 443–460 (1979).
>
> [5] Huang, Xinmeng, et al. "Uncertainty in Language Models: Assessment through Rank-Calibration" EMNLP (2024)

---

> > ### Author Response · Authors · 2025-11-26
> > **Gentle Reminder Regarding Our Rebuttal**
> >
> > We wanted to kindly follow up on our rebuttal. We are grateful for your careful and constructive feedback, which has genuinely helped strengthen our work. We recognize that reviewers carry significant responsibilities during this period and sincerely appreciate your time. Please do not hesitate to let us know if any concerns remain.

---

> > > ### Comment · Reviewer_y9ka · 2025-11-28
> > > **Reviewer Response**
> > >
> > > Thank you for the authors’ response, but I still hold my own views on two points:
> > >
> > > 1. Compared with directly refusing to answer, the model should output its confidence more accurately. From a usability perspective, if the model simply refuses, it provides no informational value. Therefore, I think metrics like AUROC are more appropriate.
> > >
> > > 2. RI itself requires a certain amount of data to estimate the necessary parameters, so its applicability may be limited.
> > >
> > > I appreciate the authors’ patient replies.

---

> > > > ### Author Response · Authors · 2025-11-28
> > > >
> > > > We appreciate the reviewer's thoughtful perspective and understand the preference for confidence scores over refusal from a usability standpoint. We' now address them point by point.
> > > >
> > > > **Why we propose RI: knowledge-aware refusal is a natural ability and should be measured.**
> > > >
> > > > We fully agree that accurate confidence scores are valuable for end users and that, when such scores are available, metrics such as AUROC are indispensable. We also do not claim that refusal mechanisms can replace fine-grained confidence estimation, which is not the goal of our paper. Our paper didn't improve the refusal ability. Rather, our goal is to **measure** refusal ability itself.
> > > >
> > > > This refusal ability, which we term *knowledge-aware refusal*, is a naturally emerging capability of modern LLMs. Knowledge-aware refusal measures whether LLMs know what they don't know, a fundamental aspect of model self-awareness[1]. Moreover, since no commercial LLMs currently provide confidence estimates for their outputs, knowledge-aware refusal is the only mechanism these models can rely on to avoid fabricating answers. We therefore believe this ability is worth measuring, and worth measuring accurately.
> > > >
> > > > RI is a metric designed specifically for evaluating knowledge-aware refusal. It does not replace but rather **complements** AUROC, as they serve different purposes:
> > > > - Measuring confidence scores or other continuous signals → Use AUROC, ECE, etc.
> > > > - Measuring LLM refusal behavior → Use Refusal Index.
> > > >
> > > > **From a utility perspective, is refusal pointless if we have good confidence estimation?**
> > > >
> > > > While discussing whether confidence scores are superior to refusal goes beyond the scope of our paper, we are happy to share our perspective. Ideally, if we had reliable confidence estimation, refusal would indeed be less informative, since confidence scores always convey more granular information. However, there are two reasons why basic LLM refusal ability remains worth investigating:
> > > >
> > > > 1. **Confidence scores are often unavailable.** When we do not explicitly prompt LLMs to generate confidence scores, refusal is their default mechanism for avoiding fabricated answers. Furthermore, when LLMs generate long-form responses, assigning confidence scores to each factual claim is impractical; the expected behavior is to generate only information grounded in the context, implicitly "refusing" to include uncertain information. In many practical scenarios, LLMs must rely on knowledge-aware refusal.
> > > >
> > > > 2. **For factual knowledge tasks, confidence scores provide limited additional value.** To illustrate this, we regenerated LLM answers on each SimpleQA question multiple times and plotted the distribution of correctness frequency.
> > > >
> > > > [[👀View the Figure]](https://cdn-uploads.huggingface.co/production/uploads/62cd3a3691d27e60db0698b0/99i6XT4TZyec3p0f5JEXD.png)
> > > >
> > > > We observe that for most questions, LLMs either consistently answer correctly or consistently answer incorrectly. This finding aligns with prior work[2]. That is, for factual tasks, LLMs either "know" the answer or "don't know" it, leaving little room for partial correctness. In this regime, an accurate confidence score would be close to either 0 or 1, providing essentially the same information as a binary refusal decision.
> > > >
> > > > **Regarding sample size requirements:**
> > > >
> > > > Thank you for raising this point. We would like to clarify that RI requires data for estimation because it is a metric, just like F1, AUROC, or refusal rate. Our experiment in Appendix H shows that RI requires approximately 1,200 samples to achieve stability comparable to other metrics like F1, only about 20% more. In practice, RI should be as practical to use as standard metrics.
> > > >
> > > > We are grateful for the reviewer's detailed and thoughtful feedback, which has helped strengthen the contributions of our paper. While we understand the preference for confidence scores from a utility standpoint, we hope the reviewer recognizes the importance of our contribution in measuring an under-explored yet fundamental capability of LLMs. We will expand the Discussion to explicitly address this complementary relationship. We would sincerely appreciate it if you could reconsider your evaluation in light of our response. Thank you!
> > > >
> > > > [1] Wei et al., 2024 – "SimpleQA: Measuring Short-form Factuality in Large Language Models"
> > > >
> > > > [2] Kadavath et al., 2022 – "Language Models (Mostly) Know What They Know"

---

### Official Review · Reviewer_hVoC · 2025-10-29

**Soundness:** 3
**Presentation:** 3
**Contribution:** 3
**Rating:** 6
**Confidence:** 3

**Summary:**

- This paper investigates knowledge-aware refusal in LLMs—the ability to refuse questions they are unlikely to answer correctly while not refusing questions they can answer. The authors identify that existing metrics like correctness conditioned on non-refusal are easily biased by refusal tendency and manipulated through system prompts.
- The authors propose the Refusal Index, which measures the Spearman correlation between refusal probability and error probability. They develop an efficient estimation procedure using Gaussian copula fitting that requires only two samples per question, making it tractable compared to expensive sampling-based calibration methods.
- The paper validates this metric by showing: (1) stability across different prompts for refusal tendencies, (2) clean correlation with sampling-based calibration methods, and (3) consistent model rankings across evaluation settings.
- Using the validated metric, the authors show: (1) cautious prompts increase refusal rates but do not improve calibration, (2) the Refusal Index is largely independent of model capability and aligns more with model family, and (3) removing or adding misleading context degrades refusal calibration.

**Strengths:**

- The Refusal Index captures something fundamental about a model's calibration and remains stable across different prompting strategies.
- Despite the mathematical complexity, the metric requires only two samples per question—nearly as cheap as computing accuracy.
- RI has strong correlation with expensive sampling-based calibration metrics
- Interesting and diverse experimental results. The finding that refusal calibration is largely independent of model capability and instead aligns with model family is particularly striking—it suggests calibration may be a distinct dimension of model quality worth optimizing separately.

**Weaknesses:**

- The mathematical presentation of RI feels dense. Figure 1 hints that the Refusal Index captures convexity of the curve in refusal rate vs. correct answer space, and further developing this intuition and/or motivating the Gaussian copula fit could aid clarity.
- Greater discussion on whether the Refusal Index measures something fundamentally different than sampling-based calibration methods, or simply serves as a more sample-efficient proxy, would be helpful. The appendix contains interesting results on sample efficiency—a direct head-to-head comparison showing how much more sample-efficient RI is than naive calibration-based metrics would better motivate its advantages.
- The stability results in Table 2 focus on Qwen and Mistral models, but Figure 4 shows Gemma-3-12b's Refusal Index varying considerably across prompts (roughly 0.1 to 0.3). It's unclear whether Gemma is an outlier or if this variation is typical. Showing both the Figure 4 analysis and Table 2 stability results on a broader, consistent set of models would clarify how stable the metric actually is in practice. This concern applies more broadly to later experiments—either evaluate more models consistently or be more intentional about which models are presented and why.
- The frontier model evaluation figure is interesting, but it's not possible to identify which specific model corresponds to each data point—only the model family is discernible, not the generation or size.

**Questions:**

- How is stability computed in Table 2? Could differences in distribution concentration affect the apparent variability of different metrics? What level of RI variation across prompts (e.g., 0.1 to 0.3 for Gemma-2-9b) should be considered acceptable? Would it be possible to show empirical data points on iso-RI curves (as in Figure 3) for more models? This would help clarify whether the Gemma-2-9b variability pattern is typical.
- The finding that RI aligns with model family rather than capability is interesting, but it's difficult to identify specific models in Figure 5. Could you provide clearer labels or a table showing individual model RI scores?
- Is RI measuring something fundamentally different from sampling-based calibration methods, or is it primarily a more sample-efficient approximation? A naive alternative would be to estimate RI through extensive sampling without Gaussian copula fitting—how would this compare?

---

> ### Author Response · Authors · 2025-11-21
>
> We thank the reviewer for the thoughtful and constructive feedback. We are encouraged that the reviewers find RI captures a fundamental aspect of calibration and appreciate our strong experiments. We carefully address your concerns point by point below.
>
> ---
>
> > **W1**: The mathematical presentation of RI feels dense. Figure 1 hints that the Refusal Index captures convexity of the curve, and further developing this intuition and/or motivating the Gaussian copula fit could aid clarity
>
> Thank you for raising this, and we agree that Figure 1 can be made more accessible to strengthen our claim. In the original manuscript, we have provided the intuition behind RI and the accuracy–refusal trade-off in Section 4.2, with more technical derivation in Appendix E. **To make the math more intuitive, in the revision, we now include references to the Discussion section in Figure 1's caption. [[🔗L069-L017]](https://openreview.net/pdf?id=9gJBhkLRat#nameddest=hVoC-W1)**
>
> ---
>
> > **W2 & Q3**: Is RI measuring something fundamentally different from sampling-based calibration methods, or is it primarily a more sample-efficient approximation?
>
> Thank you for this important question.
> Conceptually, RI and sampling-based rank-calibration target the similar underlying quantity: how well the model's refusal behavior aligns with incorrectness. The key differences are:
>
> **How refusal probability is estimated.** Sampling-based methods explicitly estimate prediction probabilities from multiple samples under temperature=1 and estimate refusal probability using refusal frequency. However, it's unknown how accurate this estimate is. In comparison, RI directly estimates the rank correlation from observable binary outcomes of refusal and error.
>
> **Efficiency.** In Section 3.3, we compare RI with AUROC computed from 100 samples per question using P(Answering) as in SimpleQA. RI achieves high correlation with this AUROC while requiring only 2 generations per question, compared to 100.
>
> We agree that a clearer presentation would be beneficial. **In the revision, we have followed your suggestion by adding a table that directly compares RI with other calibration methods to better highlight RI's advantages. [[🔗L1296-L1312]](https://openreview.net/pdf?id=9gJBhkLRat#nameddest=hVoC-W2/Q3)** Please see the table below:
>
> | Method                   | Typical calibration method(s)                       | Unbiased to true refusal probability? | Computational cost                                                    |
> | ------------------------ | --------------------------------------------------- | ------------------------------------- | --------------------------------------------------------------------- |
> | Linear Probe             | Train a linear classifier on internal hidden states | ✗                                     | $S*N_\text{train}*d$ Probe training + $N$ generations + $N$ inferences                 |
> | Black-box Estimator      | Train an auxiliary classifier on output text        | ✗                                     | Calibrator training on $N_\text{train}$ samples + $N$ generations + $N$ inferences |
> | Verbalized Confidence    | Ask model to output a numeric confidence value      | ✗                                     | $N$ generations + $N$ confidence-score generations                        |
> | Sampling-based           | Use refusal frequency to approximate refusal prob.  | ✓                                     | $S*N$ generations (S: samples per question)                         |
> | **Refusal Index (ours)** | Two-pass evaluation, no extra model                 | ✓                                     | $2N$ generations                                                    |

---

> ### Author Response · Authors · 2025-11-21
>
> > **W3**: Gemma-3-12b's Refusal Index varies considerably across prompts (roughly 0.1–0.3). Is Gemma an outlier?
>
> Thank you for pointing this out. In our experiments, Gemma3-12B behaved as an outlier, with less consistent adherence to the system instruction than the other models. This instability **is not specific to the cross-prompt settings in Table 2, but also appears in single-prompt settings** (see single prompt results from SimpleQA in Appendix F).
>
> Therefore, we did not include it in the cross-prompt stability test initially. We hypothesize that this may be related to its smaller model size and corresponding limitations in following system prompts consistently, which we have listed as one of the failure modes of RI in Appendix A.
>
> We agree that showing results for more models would strengthen our claims. **In the revision, we have updated the table to include results for all models tested. [[🔗L294-L307]](https://openreview.net/pdf?id=9gJBhkLRat#nameddest=hVoC-W3)** While Gemma3 shows a higher coefficient of variation for RI than other models, RI still exhibits the lowest variance compared to other metrics.
>
> | Type | Model | Accuracy | Refusal | C/A | F-score | Weighted | RI |
> | :--- | :--- | :---: | :---: | :---: | :---: | :---: | :---: |
> | **Normalized Difference ($\Delta$)** | Mistral-123B | -0.40 | +0.93 | +0.37 | -0.16 | -0.83 | **+0.06** |
> | | Qwen2-35B | -0.47 | +0.95 | +0.12 | -0.31 | -0.62 | **-0.19** |
> | | Qwen2.5-72B | -0.84 | +0.43 | +0.50 | -0.60 | -1.32 | **-0.07** |
> | | Qwen3-32B | -0.96 | +0.54 | +0.48 | -0.71 | -1.42 | **+0.14** |
> | | Gemma-3-12B | -1.31 | +2.04 | +0.96 | -0.93 | +1.79 | **+0.42** |
> | | Average | -0.80 | +0.98 | +0.49 | -0.54 | -0.48 | **+0.07** |
> | **Coefficient of Variation ($CV$)** | Mistral-123B | 0.16 | 0.35 | 0.14 | 0.06 | 0.31 | **0.04** |
> | | Qwen2-35B | 0.22 | 0.47 | 0.06 | 0.14 | 0.32 | **0.09** |
> | | Qwen2.5-72B | 0.35 | 0.17 | 0.19 | 0.26 | 0.53 | **0.03** |
> | | Qwen3-32B | 0.35 | 0.19 | 0.17 | 0.28 | 0.51 | **0.07** |
> | | Gemma-3-12B | 0.49 | 0.76 | 0.39 | 0.36 | 0.66 | **0.23** |
> | | Average | 0.31 | 0.39 | 0.19 | 0.22 | 0.47 | **0.09** |
>
> ---
>
> > **W4 & Q2**: It's not possible to identify which specific model corresponds to each data point in the frontier model evaluation figure.
>
> Thank you for this valuable suggestion. In the original manuscript, we have included all raw results from SimpleQA in Appendix F. To make the figures easier to interpret, **in the revision, we have followed your suggestion and updated Figure 5 by adding name labels to each data point. [[🔗L442-L457]](https://openreview.net/pdf?id=9gJBhkLRat#nameddest=hVoC-W4/Q2)**
>
> ---
>
> > **Q1.1**: How is stability computed in Table 2?
>
> Thank you for asking us to clarify this. In Table 2, we report two complementary stability measures across the four refusal prompts for each model:
>
> - The **normalized difference**
>
>     $$
>     \Delta \text{Metric} = \frac{\text{Metric max} - \text{Metric min}}{\lvert \text{Metric}_\text{mean} \rvert}
>     $$
>
>     which captures how far the most and least stable runs deviate relative to the average level of the metric.
>
> - The **coefficient of variation (CV)**
>
>     $$
>     \text{CV} = \frac{\text{standard deviation}}{\lvert \text{mean} \rvert}
>     $$
>
>     which measures relative dispersion around the mean and is standard for comparing variability across metrics with different scales. **In the revision, we have updated Section 3.2 to include these formulas.[[🔗L308-L318]](https://openreview.net/pdf?id=9gJBhkLRat#nameddest=hVoC-Q1.1)**
>
> ---
>
> > **Q1.1a** Could differences in distribution concentration affect the apparent variability of different metrics?
>
> Thank you for raising this question. Both metrics are explicitly **scale-normalized**, which controls for differences in how concentrated the underlying distributions are.

---

> ### Author Response · Authors · 2025-11-21
>
> > **Q1.2**: What level of RI variation across prompts should be considered acceptable?
>
> Thank you for raising this matter. While our main claim is that RI shows much less variance than other metrics, we don't rely on cross-prompt variance when using RI for evaluation: when comparing RI across different models, we evaluate those models under the same system prompt, so variance across prompts doesn't affect the comparison. For variance within a single prompt, in Appendix C, we provide ablation experiments on how many samples are needed for acceptable variance (CV < 10%): around 1200 samples, 20% more than traditional metrics
>
> ---
>
> > **Q1.3**: Would it be possible to show empirical data points on iso-RI curves (as in Figure 3) for more models?
>
> Yes, and we agree with you that this is a very helpful visualization. **In the revision, we have extended the iso-RI visualization to include more models, including Gemma-3-12B, Qwen3-32B, Qwen3-32B-Think, Qwen2.5-72B, DeepSeek-V3, and Mistral-123B [[🔗L347-L349]](https://openreview.net/pdf?id=9gJBhkLRat#nameddest=hVoC-Q1.3)**. For each model, we plot the empirical (refusal rate, correct answer rate) pairs induced by the four prompts. This addition makes it clear which models have tightly clustered points that align with a single iso-RI curve and which models show more variation. Please see the figure below.
>
> [[🌐 View Iso_RI Plot with Different Models]](https://cdn-uploads.huggingface.co/production/uploads/62cd3a3691d27e60db0698b0/466Xck-WTlF9uGrco01RP.png)
>
> ---
>
> > **Q1.4** How stable is RI in practice?
>
> Thank you for the important question.
> We understand your concerns about whether RI's stability generalizes to all models. While our experiments have shown RI is stable across different refusal rates, RI's efficacy does not necessarily come from this stability.
>
> - Our core contribution is designing RI in a theory-driven way: RI is defined as the Spearman correlation between refusals and errors, and we use Gaussian Copula to estimate it from two-pass observations on refusal and correctness outcomes. This principled approach makes RI an efficient and faithful measurement of knowledge-aware refusal.
> - Our interesting discovery is that in practice, models return the same RI even when they exhibit different refusal rates. This stability strongly suggests that RI captures a more robust property than simply refusal rates. What's more, the fact that this property is more correlated with model family than model scale hints that knowledge-aware refusal may be a distinct dimension of model quality worth looking into.
>
> We hope these clarifications and the additional experimental evidence address your concerns. Please let us know if you have any further questions!

---

> > ### Author Response · Authors · 2025-11-26
> > **Gentle Reminder Regarding Our Rebuttal**
> >
> > We would like to kindly follow up on our rebuttal. Your careful and constructive review has genuinely helped us improve our work, and we are grateful for your efforts. We understand the demands on reviewers during this period and thank you for your continued attention. Please feel free to reach out if any questions remain.

---

### Official Review · Reviewer_Zb1L · 2025-10-31

**Soundness:** 3
**Presentation:** 3
**Contribution:** 3
**Rating:** 6
**Confidence:** 4

**Summary:**

In the blackbox approach to factual knowledge, the authors point out that current methods are limited. They will mostly look at the rejection rate and/or correct answer rate. Here, the authors propose to check whether those two things happen together, beyond chance. This method additionally avoids a lot of sampling which is often relied on for similar approaches. They provide extensive empirical testing on different models and datasets, and discuss model variations. They also check effect of prompt variation.

**Strengths:**

Reduces a true gap in a very empiric field. As an empirical study it is quite solid, checking a variety of models, datasets, but also checking the effect of prompt variation to some extent.

The empirical section is not only quantitatively very strong, but actually makes good use of it's data. Datasets and models are not only listed, but are used to make compelling arguments on different effects. Figures are clear and very understandable. Multiple appendices make the effort of testing many variations of the setup to ensure it is correctly validated.

**Weaknesses:**

An issue I find generally in the blackbox tradition of model factual knowledge is that a lot of definitions are arbitrary.
I would worry that these poorly defined targets of "checking if a model knows" or "checking if a model answers when it knows" are moving goalposts which lead to incremental progress to evaluate models which purposedly (hence blackbox) do not provide the required information to properly move forward.

Were this a journal paper in a major venue, I would ask to rework the definition of "knowledge". As this is a conference paper in a major venue I can only notify that this definition is very shaky (we are not discussing observable correct answer for a factual question, but a more interesting but very abstract notion of "knowing" a fact). I nonetheless acknowledge that this paper is taking a step in the right direction by decoupling model behaviour of refusal to answer from model knowledge, and this is why I've set a positive score.
I remain nonetheless worried that definitions are not well set - much like in previous empirical works, there is no clear gold standard for "knowing". Reasoning then becomes somewhat circular - we empirically define knowing as RI, and then show that it is better than previous methods which had set different definitions.

Along the same line the authors criticize the notion of proxy metrics. I did not understand how this new RI method is not a proxy metric, even if a better one.

On a much less important note, I've noted moments where I was confused reading. Should they seem personal, feel free to ignore them.
* Line 015/016 : "simple refusal based metrics are biased by refusal rates and yield inconsistent scores when models exhibit different refusal tendencies" --> confusing
* starting L050: I was confused again by the third paragraph of the introduction.

For both of those I only understood what was going on from the examples in l110 onwards which made your point as well as the difference between refusal based metric, refusal bias, refusal tendencies, and refusal itself as a concept much clearer. I would advise either clarifying earlier, or rephrasing.

* 024 "RI accurately quantifies a model's intrinsic knowledge-aware refusal rates capability in factual tasks." --> intrinsic knowledge aware refusal rates is not defined later in the paper, and confusing here

**Questions:**

1) could you please re-explain why you consider calibration a proxy, and not RI?

2) L151/152 "While overall refusal rates can be adjusted through input context or preference learning, the discriminative capability for knowledge-aware refusal remains more robust and consistent" - I am confused by this statement. How can changing refusal rates be more consistent than discriminating? more consistent for what? I think I understand your point that refusal rates are not the only thing we want to act on - but I don't think RI as a metric is acting on anything.

3) more of a personnal curiosity point: is there a reason why you are using Spearman rank correlation rather than another?

---

> ### Author Response · Authors · 2025-11-21
>
> We thank the reviewer for the thoughtful and constructive feedback. We appreciate that you find our empirical analysis solid and recognize that the paper addresses an important gap in evaluating knowledge-aware refusal. Below, we carefully address each of your concerns.
>
> ---
>
> > **W1**: An issue I find generally in the blackbox tradition of model factual knowledge is that a lot of definitions are arbitrary. Moving goalposts of the definition of "knowing" makes RI definition circular - we empirically define knowing as RI, and then show that it is better than previous methods which had set different definitions.
>
> Thank you for raising this important point. We fully agree that defining "knowledge" for black-box LLMs is inherently challenging, and that prior empirical works often rely on loosely defined notions. RI aims to address this gap by being more explicit about what it measures.
>
> **Definition of Knowledge in Our Paper**. In our work, an LLM "knows" a fact when it answers correctly. While this is not the only possible definition, it is the most observable one in black-box settings. To make this more explicit, **we have now clarified our definition of knowledge at the beginning of the Background section in the revision. [[🔗L095-L096]](https://openreview.net/pdf?id=9gJBhkLRat#nameddest=Zb1L-W1)**
>
> **What RI Measures**. RI does not implicitly redefine "knowing" but instead evaluates whether refusal behavior aligns with observed correctness across questions. By contrast, existing approaches often lack a clear formalization of what they measure, making it difficult to interpret their scores consistently.
>
> ---
>
> > **W2 / Q1**: I did not understand how this new RI method is not a proxy metric, even if a better one
>
> This is an excellent question. We would like to clarify why other calibration methods are proxies while RI is not. In short, to measure knowledge-aware refusal: **the correlation between refusal decisions and correctness**, previous calibration methods replace actual refusal observations with noisy, indirect scores (e.g., model-predicted confidence scores), but these **proxy scores** are not reliable predictors of model refusal decisions. To make this clearer:
>
> 1. **What is needed for evaluating knowledge-aware refusals**. We assess knowledge-aware refusals by measuring the correlation between refusal and correctness. A higher correlation means the model is more likely to refuse when it is likely wrong, and more likely to answer when it can provide a correct answer (i.e., it "knows" the answer).
> 2. **Why calibration methods are proxies**. Calibration methods do not actually observe refusal decisions. Instead, they use a "proxy" model to approximate refusal probability—typically by instructing the model to state its own confidence score or by training another model to predict confidence. As empirical evidence shows, these proxy scores can be heavily biased and imperfect for predicting the model's actual refusal decisions (We have provided more evidence in the revision).
> 3. **Why RI is not a proxy metric**. RI directly derives the correlation from observable refusal decisions and correctness alone. We call RI a direct measure because it does not assume how the model works internally and does not use an imperfect, proxied predictor of refusal decisions.
>
> **In the revision, we have added a new Background section that provides a more concrete explanation of calibration metrics' limitations in terms of their proxy nature. [[🔗L128-161]](https://openreview.net/pdf?id=9gJBhkLRat#nameddest=Zb1L-W2/Q1)**
>
> ---
>
> > **W3**: I've noted moments where I was confused reading
>
> Thank you for identifying these ambiguous points. We understand that these sentences might be compact. **In the revision, we have moved these descriptions to the Background section to explain them in more detail.**
>
> > **W3**: Intrinsic knowledge aware refusal rates is not defined later in the paper, and confusing here
>
> Thank you for pointing this out. Here "intrinsic" only indicates that the capability is inherent and does not change the meaning of knowledge-aware refusal. **In the revision, we have updated the abstract to simplify this expression and avoid confusion.[[🔗L022]](https://openreview.net/pdf?id=9gJBhkLRat#nameddest=Zb1L-W3)**

---

> ### Author Response · Authors · 2025-11-21
>
> > **Q2: Why discriminative capability is more consistent than refusal rates**
>
> Thank you for asking for clarification.
>
> Discriminative capability refers to how well a model distinguishes harder questions from easier ones based on its refusal behavior. This differs from the absolute refusal rate. A model's refusal rate can be trivially altered by prompting or preference tuning, as shown in prior work and in our experiments. However, such interventions do not inherently improve the model's ability to *rank* questions by difficulty. The discriminative property remains stable across such changes. RI isolates exactly this invariant property by focusing on rank correlation. **In the revision, we have updated the explanations following Eq. 2 to make this distinction clearer. [[🔗L192-L194]](https://openreview.net/pdf?id=9gJBhkLRat#nameddest=Zb1L-Q2)**
>
> ---
>
> > **Q4: Why Spearman rank correlation**
>
> Good question. The use of Spearman correlation follows from the Gaussian copula formulation we use to model the dependence between refusal and error probabilities. Under the Gaussian copula, the Pearson correlation of the latent variables has a closed-form, one-to-one mapping to Spearman correlation, which makes estimation more stable and interpretation more meaningful. Furthermore, any monotone transformation of this dependence structure leads to an equivalent metric, so Spearman is a natural choice.
>
> We again thank the reviewer for the thoughtful comments. We hope these clarifications and revisions address the concerns raised. Please let us know if you have any further questions!

---

> > ### Author Response · Authors · 2025-11-26
> > **Gentle Reminder Regarding Our Rebuttal**
> >
> > We wanted to kindly follow up on our rebuttal. We appreciate your careful and constructive review, which has genuinely helped improve our work. We understand reviewers have many responsibilities during this period and thank you for your time. Please do not hesitate to reach out if any concerns remain unaddressed.

---

### Meta-Review · Area_Chair_Hvna · 2026-01-06

**Summary:**

The paper introduces a metric Refusal index (RI) for measuring knowledge based refusals which is more stable compared to previous e.g. calibration based refusal measures. The reviewers raised some concerns regarding the motivation for such a metric and its advantages compared to prior works but I believe these questions were sufficiently addressed in the rebuttal.

**Reviewer Concerns:**

See below.

**Reviewer Scores:**

Reviewer Zb1L raises a concern regarding the definitions of LLM’s knowledge and usefulness of various proxy metrics in refusals, the authors respond to these questions, and I believe the reviewer would retain the score 6.

Reviewer hVoC raises concerns related to deeper understanding of RI index metric and additional results on stability. I believe the authors sufficiently addressed these questions and the reviewer would retain score 6.

Reviewer y9ka raises several concerns: 1) lack of background on shortcomings of prior works and choice of baselines, 2) advantages of RI compared to AUROC. I believe the authors’ rebuttal sufficiently responded to these concerns such as providing enough motivation for output-based refusal metrics compared to calibration threshold based metrics, however the reviewer decided to retain their score 4.

Reviewer 5y9a raised technical design questions (e.g. using bivariate Gaussian distribution and Spearman rank correlation in RI). I believe the authors answered these questions in the rebuttal and the reviewer should have raised the score 2->4 or 2->6.

Overall, I believe this work makes an interesting and novel contribution to the LLM refusal evaluation area and the reviewers didn’t raise any major concerns.

---

### Decision · Program_Chairs · 2026-01-26

Accept (Poster)